# Self-supervised video pretraining yields robust and more human-aligned visual representations

**Nikhil Parthasarathy**[†]    **S. M. Ali Eslami**    **João Carreira**    **Olivier J. Hénaff**
Google DeepMind

## Abstract

Humans learn powerful representations of objects and scenes by observing how they evolve over time. Yet, outside of specific tasks that require explicit temporal understanding, static image pretraining remains the dominant paradigm for learning visual foundation models. We question this mismatch, and ask whether video pretraining can yield visual representations that bear the hallmarks of human perception: generalisation across tasks, robustness to perturbations, and consistency with human judgements. To that end we propose a novel procedure for curating videos, and develop a contrastive framework which learns from the complex transformations therein. This simple paradigm for distilling knowledge from videos, called VITO, yields general representations that far outperform prior video pretraining methods on image understanding tasks, and image pretraining methods on video understanding tasks. Moreover, VITO representations are significantly more robust to natural and synthetic deformations than image-, video-, and adversarially-trained ones. Finally, VITO's predictions are strongly aligned with human judgements, surpassing models that were specifically trained for that purpose. Together, these results suggest that video pretraining could be a simple way of learning unified, robust, and human-aligned representations of the visual world.

## 1 Introduction

With the explosion of recent AI breakthroughs, humans now interact with and depend on the outputs of these models at an unprecedented rate. It is therefore increasingly important that these models be aligned with human abilities, judgements, and preferences. In the context of computer vision systems, human alignment can be quantified with accurate generalization across a wide range of tasks [1–3], robustness to various input deformations [4], and consistency with human perceptual judgements [5]. While each of these challenges has been tackled separately, progress along one axis has often come at the expense of the others. For example, gains in robustness [6] or temporal understanding [7–9] have thus far come at the cost of spatial understanding, and scaling the model and dataset size, while improving task-generality and robustness [10, 11], can be detrimental for their consistency with human perception [11, 12].

In this work we question this trend, and ask whether improvements to all aspects of human alignment can be made with the appropriate pretraining methodology. Specifically, humans and animals have long been thought to learn from the dynamic evolution of natural scenes [13–15] and we hypothesize that artificial visual systems will be more aligned by appropriately leveraging natural video pretraining. In particular, while many current self-supervised methods [16–19] learn representations that are invariant to synthetic augmentations that capture important image priors such as scale-, color-, and translation-invariance, these represent a small part of the complex (and signal-rich) changes in pose, viewpoint, and motion that are captured from natural videos. Predicting the evolution of videos is also a natural means of learning intuitive physics and model-based reasoning [20–22].

---

[†]Current affiliation: NYU Center for Neural Science, work done while interning at DeepMind.

37th Conference on Neural Information Processing Systems (NeurIPS 2023).

Practically, we develop a self-supervised contrastive framework which learns to locate the most stable and distinctive elements in temporally displaced video frames, and maximizes their invariance. Secondly, we find the statistics of standard video datasets to have a detrimental effect on the quality of the resulting representations, as measured by their performance on canonical scene understanding tasks. We therefore introduce a simple, yet powerful video curation procedure—VideoNet—which aligns their class distribution with that of ImageNet, and which redresses the imbalance between image and video learning. In concert, this paradigm constitues a new methodology for distilling the knowledge of **vi**deos in**to** visual representations: VITO.

VITO yields task-general representations that perform well across both spatial and temporal understanding tasks. Particularly, VITO shows large gains over prior video pretraining efforts in scene understanding tasks, while achieving similarly large performance gains over image pretraining on video understanding tasks. Furthermore, VITO significantly outperforms the default ImageNet pretraining as well as adversarial pretraining on image classification tasks subject to natural distribution shifts. Finally, we find that even without a significant expansion in model size, VITO is not only task-general and robust in performance, but also quantitatively captures multiple aspects of human perceptual judgements, surpassing models specifically trained for that purpose.

## 2   Related work

**Learning general visual representations from videos.** Many prior works have considered self-supervised representation learning for capturing spatio-temporal invariances, beginning with methods that leveraged temporal coherence, optical flow, and object tracking [23–31]. More recently, many successful approaches have leveraged contrastive learning, masked autoencoding, and other self-supervised pretext tasks to learn strong video representations [32–38]. However, most of these methods employ specialized video architectures and only transfer to video-based tasks such as action recognition and motion segmentation.

Yet natural motion-induced deformations are powerful learning signals that should allow for learning better *image* representations as well. Indeed, human infants can form complex understanding of objects and shape within months, specifically driven by their observations of how they move [39, 40]. Given this inspiration, some works have demonstrated that self-supervised contrastive learning in videos can lead to aspects of efficient human learning and robust recognition [41–43]. In computer vision, cycle-consistency [44, 45] and optical flow [46, 47] have been used to learn correspondences between temporally ordered image patches. The most similar works to ours utilize video-based contrastive learning [7–9] to improve performance on temporal understanding tasks, however they do so at the cost of spatial scene understanding.

**Robustness to distribution shifts.** As standard benchmarks have been progressively saturated [48], the community has turned to measuring robustness to adversarial attacks [49], corruptions [50], and out-of-distribution datasets [4, 51–53]. We focus on a subset of these benchmarks that are as "natural" as possible, to evaluate generalization with respect to shifts that are most likely to appear in the real world. While there have been many efforts to specifically encourage regularize models for these kinds of robustness [54–57], we instead investigate the complementary question of whether image and video pretraining differ in this respect.

**Human-aligned representations.** Most recent progress in achieving more behaviorally-matched representations has been by scaling existing approaches. Indeed, recent examples [10, 11, 58] show that as data and model sizes grow by orders of magnitude, generality and robustness of representations tend to emerge. Moreover some aspects of human perception such as an increased shape-bias and consistency with human perceptual behavior [11, 59] can be captured reasonably well by certain large models. However this scaling property tends to be brittle, with some large-scale models displaying significantly worse consistency with human perception [11, 12]. Additionally, more recent work on alignment has found that scaling and architecture are not as important for alignment on specific benchmarks, in comparison to the training dataset and objective function [60]. Therefore, while scaling may continue to lead to task-performance gains, it is unclear whether only scaling image-based pretraining will close the gap with general human behavior. We therefore explore the complementary and potentially synergistic question of whether video pretraining can improve the task-generality, robustness, and behavioral similarity of learned visual representations.

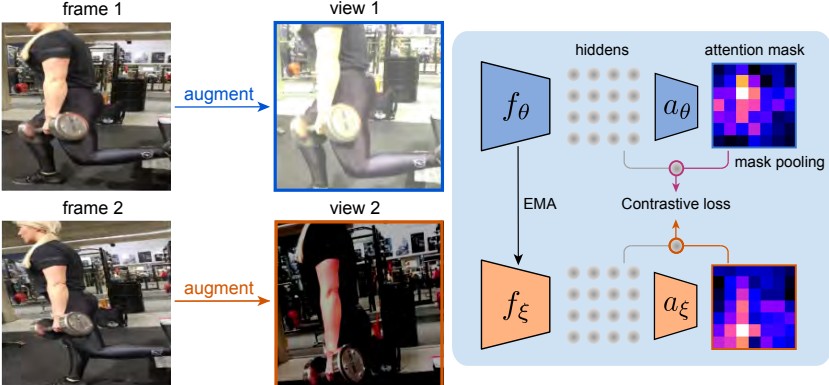

**Figure 1:** Learning to attend to related video content. Each augmented frame is encoded by the network $f$ as a spatial array of hidden vectors. The attention module $a$ takes as input features from one view and produces a mask that isolates features that are likely to be predictive of the other, temporally-displaced view. The attention-gated features are pooled accordingly, and both the feature extractor and attention module are trained to satisfy the contrastive objective. Subscripts $\theta$ and $\xi$ refer to online and target (EMA) networks respectively.

## 3 Method

We pretrain image representations using video datasets, then transfer them to a range of downstream tasks that test image, video, and robust understanding. We adopt the ResNet-50 architecture for our initial exploration, then validate our results with Swin transformers (see Sec. B.4).

### 3.1 Self-supervised pretraining

Our method for distilling **vi**deos in**to** image representations, **VITO**, builds robust visual representations by learning to track stable and distinctive content in videos while they evolve over time.

**Natural video pipeline.** The key to our method is to distill the natural transformations present in videos into image-based representations. Given a video-clip, we sample frames according to a distribution $\mathcal{T}$ and further transform each frame with image-based augmentations:

$$\boldsymbol{v}^1 \sim \mathcal{A}_1(\boldsymbol{x}_1) \qquad \boldsymbol{v}^2 \sim \mathcal{A}_2(\boldsymbol{x}_2) \qquad \boldsymbol{x}_1, \boldsymbol{x}_2 \sim \mathcal{T}(\{\boldsymbol{x}_t\}_{t=1,\dots,T}) \tag{1}$$

where the distribution $\mathcal{T}$ samples frames uniformly from a video clip of length $T = 2.56s$ and the image transformations $\mathcal{A}_l$ include random cropping, flipping, blurring, and point-wise color transformations [61], see appendices A.1 and B.2, and Figure B.3 for an ablation.

We note that video frames (or even uncurated image data) typically differ from the statistics of (object centered) ImageNet images, with more variable viewpoints and a larger field-of-view that can cover multiple objects in complex scenes. As a result, the aggressive random cropping from [61] (whose smallest crops cover only 8% of the original image) can result in "positive" pairs with very different semantic content (e.g. entirely different objects). We therefore suggest and empirically validate that larger crop sizes (e.g. increasing the minimum crop size to 40%) are beneficial when learning from real-world video frames (see Figure B.2).

**Multi-scale contrastive attention pooling.** Standard contrastive frameworks use global average pooling of hidden vectors to obtain a single representation of each view. It has been shown that using dense contrastive losses can lead to significant improvements [62–65], but these methods require establishing correspondences across views. Whereas correspondences can easily be obtained from static images, when temporal deformations are introduced they require some form of object or point tracking [46]. Furthermore, with the larger field-of-view of video frames, correspondence learning becomes an increasingly difficult task. In this work, we propose a more general, adaptive method for learning correspondences at multiple scales. Our method learns what features should be attended to in order to solve the contrastive learning problem across temporally displaced views.

As shown in Figure 1, given a view $\boldsymbol{v}^l$ the feature extractor outputs a spatial map of feature vectors $\boldsymbol{h}_\theta^{l,s} \in \mathcal{R}^{h \times w \times c}$ at a given scale $s$, where different scales correspond to the outputs of different blocks

of a ResNet for example. At each scale, we introduce a 2-layer attention MLP $a_\theta^s$ which outputs a mask $\boldsymbol{m}^{l,s} = \text{softmax}(a_\theta(\boldsymbol{h}_\theta^{l,s}))$ that we use to spatially weight and pool hidden vectors:

$$\hat{\boldsymbol{h}}_\theta^{l,s} = \sum_{i,j} \boldsymbol{m}^{l,s}[i,j] \, \boldsymbol{h}_\theta^{l,s}[i,j] \tag{2}$$

which we we concatenate and transform with the two-layer MLP projector: $\boldsymbol{z}_\theta^l = g_\theta(\hat{\boldsymbol{h}}_\theta^l)$ where $\hat{\boldsymbol{h}}_\theta^l = [\hat{\boldsymbol{h}}_\theta^{l,s}, \ s \in 1...S]$. In our experiments, we find that for the canonical ResNet-50 architecture, attending over the outputs of the last two ResNet blocks (i.e. $S = 2$) is optimal given our evaluations. These hidden vectors are then transformed with a standard two-layer MLP $g_\theta$, yielding projections $\boldsymbol{z}_\theta^l = g_\theta(\hat{\boldsymbol{h}}_\theta^l)$. We enforce invariance across views using the standard InfoNCE loss [66], encoding targets with slowly-varying *target* networks $f_\xi$ and $g_\xi$ that are exponential moving averages of the online network [61]

$$\mathcal{L}^{ij}(\theta;\xi) = -\log \frac{\exp(\boldsymbol{z}_\theta^i \cdot \boldsymbol{z}_\xi^j)}{\exp(\boldsymbol{z}_\theta^i \cdot \boldsymbol{z}_\xi^j) + \sum_n \exp(\boldsymbol{z}_\theta^i \cdot \boldsymbol{z}_\xi^n)}. \tag{3}$$

$\{\boldsymbol{z}_\xi^n\}_n$ are *negative* features computed from frames from other videos in the batch. The final, multi-view loss is evaluated for all pairs $\mathcal{L}(\theta;\xi) = \sum_{i \neq j} \mathcal{L}^{ij}(\theta;\xi)$.

### 3.2 Addressing dataset domain mismatch

We began investigating the potential for learning general representations from videos, using standard datasets including Kinetics, AudioSet, and YouTube-8M. However, Kinetics is quite small and is limited in scope to human actions. On the other-hand, AudioSet and YouTube-8M are noisy and have very imbalanced class distributions. Additionally, prior work has shown that even self-supervised methods are quite sensitive to the pretraining distribution [67]. Yet over the last decade, it has been shown that ImageNet can be used for learning image representations that transfer well to many downstream tasks. As a result, we hypothesized that collecting a minimally-curated video dataset matched to the rough properties of ImageNet would be beneficial for learning a more general visual model from videos.

To test of this hypothesis, we developed a data curation pipeline—*VideoNet*—to filter online videos such that our training data more closely matches the distribution of ImageNet categories. For each of the 1,000 ImageNet categories, we retrieved 5,000 video clips whose title included the category's name or a synonym. We then filtered these videos by applying an image classifier (pretrained ResNet-50 on ImageNet) to verify that the videos contained the intended object category. We classified the first 100 frames of each video and discarded videos for which the query category was not equal to the ResNet's top-1 prediction for any of the frames. We also discarded videos of less than $10s$ in length.

While the VideoNet procedure is close in conceptualization to the method used to create the R2V2 dataset proposed by Gordon et al. [7], it differs in a few ways. First, we utilize full video clips that allow us to uniformly sample frames at any time point rather than the fixed sampling of frames that are $5s$ apart in R2V2. Second, by using the ImageNet classifier to filter videos, we can reduce mismatch with the ImageNet distribution that can arise from incorrect tagging and noisy labeling of online videos. This is verified by the fact that only 1.18M of the 5M retrieved videos met our filtering criteria. We also note that the use of classification-based filtering is just one method of curation. While we demonstrate in Sec. 4.3, that this curation does provide large benefits in the context of video pre-training compared with existing datasets, there is still great potential to make improvements by utilizing larger target datasets (such as ImageNet-22K) and utilizing alternative curation strategies such as the nearest-neighbor retrieval proposed by [10] in creating the LVD-142M image dataset.

## 4 Results

Humans are able to solve a range of visual tasks that require complex spatial and temporal reasoning, including generalizing to noisy or out-of-distribution (OOD) scenarios. Therefore, we first benchmark VITO against image and video pretrained models on a variety of tasks to demonstrate sufficient generality and robustness in task performance. We then assess whether VITO not only captures these task-based properties, but also displays strong quantitative alignment with human behavior.

## 4.1 VITO generalizes across diverse visual tasks

We present in Table 1 the transfer performance of VITO compared to strong supervised and self-supervised baselines on dense scene understanding (semantic segmentation and object detection), video understanding (video segmentation and action recognition), and out-of-distribution (OOD) object recognition. On every benchmark, VITO either outperforms or is competitive with the best baselines *for that specific task*.

| Pretraining | Dataset | Scene Understanding | | Video Understanding | | OOD Recognition | |
|---|---|---|---|---|---|---|---|
| | | ADE20K (mIoU) | COCO (mAP) | DAVIS ($\mathcal{J}\&\mathcal{F}$ mean) | UCF101 (top-1) | IN-A (top-1) | IN-Vid (pm0/ pm10) |
| Random | - | 27.9 | 39.0 | - | - | - | - |
| *Standard image pretraining* | | | | | | | |
| Supervised | ImageNet | 33.5 | 44.2 | 66.1 | 83.4 | 2.2 | 67.7/52.4 |
| BYOL [61] | ImageNet | 38.8 | 43.7 | 66.6 | 85.6 | - | - |
| MoCLR [68] | ImageNet | 39.2 | 43.9 | 65.5 | 85.5 | 3.7 | 64.7/50.0 |
| DINO [19] | ImageNet | 39.0 | **44.3** | 65.3 | 85.4 | 5.0 | 65.2/52.0 |
| *Robust image pretraining* | | | | | | | |
| Stylized-IN [56] | SIN+IN | - | - | - | 83.3 | 2.0 | 68.4/51.7 |
| L2-Robust [54] | ImageNet | - | - | - | 83.7 | 2.1 | 65.2/51.6 |
| *Video pretraining* | | | | | | | |
| VIVI [69] | YT8M | 34.2 | 41.3 | - | - | 0.5 | 57.9/36.5 |
| MMV-VA [70] | AS + HT | 32.5 | 41.3 | - | - | - | - |
| VINCE [7] | R2V2 | 35.7 | 42.4 | 66.1 | - | - | - |
| VFS [8] | K400 | 31.4 | 41.6 | 67.8 | - | - | - |
| CycleCon [9] | R2V2 | 35.6 | 42.8 | - | 82.8 | 0.4 | 50.4/30.1 |
| VITO | VideoNet | **39.4** | 44.0 | **68.2** | **87.4** | **5.4** | **70.6/57.2** |

**Table 1:** VITO representations generalize to a variety of tasks in both image and video modalities, surpassing models specialized for each task. For external models, we finetune publicly available checkpoints.

**Scene understanding.** We first note that VITO provides large gains over all prior video pretraining methods on scene understanding and robust object recognition. We further validate these comparisons on three additional benchmarks and find that VITO strongly outperforms the prior work across all 5 datasets (PASCAL/ADE20K/COCO/LVIS/IN-1K, see Table B.3). For example, VITO improves over VIVI [69] by 2-10%, highlighting the importance of data curation and our contrastive formulation. VITO improves over VINCE [7] by 1-12%, highlighting the importance of fine-grained temporal deformations. Finally, VITO improves even over MMV [70] by 2-15%, despite their use of large-scale text supervision, highlighting the relevance of video-only learning.

Compared with the best supervised and self-supervised image-pretraining methods VITO achieves competitive performance on these same benchmarks (Table 1 and Table B.3). To our knowledge, VITO is the first video pretrained method to close the gap with ImageNet pretraining on large-scale scene understanding benchmarks such as these.

**Video understanding.** We next ask whether this increased spatial understanding come at the cost of traditional benefits of video pretraining on video tasks. We find that this is not the case, evaluating on DAVIS segmentation and UCF-101 action recognition. On DAVIS, which tests the ability to segment an object over its dynamic temporal evolution, VITO features capture fine-grained temporal deformations of objects far better than ImageNet pretraining methods, as well as the best video pretraining methods (See Table B.4 for additional comparisons). On UCF-101, which tests the ability to classify global spatio-temporal features, we find that a simple average pooling of VITO frame representations again outperforms all image pretraining and prior frame-based video pretraining significantly. VITO even outperforms a number of recent methods that use specialized video architectures (See Table B.5). While VITO under-performs relative to the best video models, we note that these methods either cannot be tested or under-perform on spatial understanding. Additionally, as shown in Table B.5 and Sec. A.5, simple learned temporal pooling strategies on top of VITO representations further close the gap with the best video architectures.

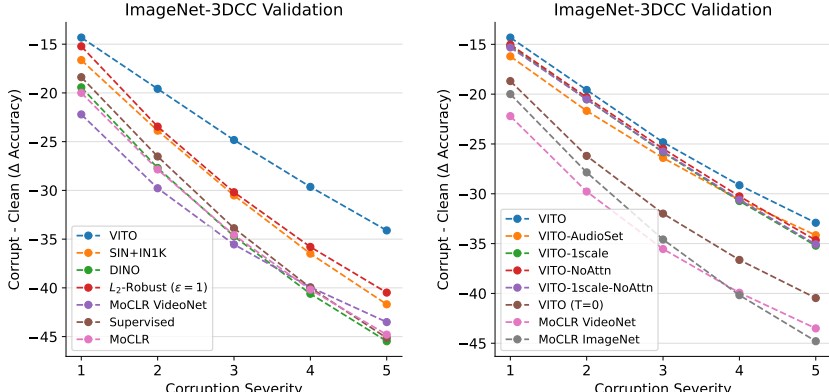

**Figure 2:** ImageNet-3DCC validation accuracy for different levels of corruption severity. (Left): Comparisons with prior work including methods specifically designed to enhance robustness (SIN+IN1K and L2-Robust). (Right): comparisons with ablations of the VITO method/model.

**Object recognition under distribution shifts.** A key feature of human perception is being able to generalize under distribution shifts away from the training data. The standard ImageNet benchmark does not test this, as the validation set is drawn from a similar distribution as the train set. We hypothesize that while ImageNet pretraining can lead to strong performance in-distribution, pretraining on videos can endow models with better generalization capabilities.

We thus evaluate on a suite of benchmarks designed to test distributional robustness [4]. To test recognition under *natural* shifts we evaluate on the ImageNet-Vid-Robust and ImageNet-A benchmarks (Table 1). ImageNet-Vid-Robust tests generalization of image classifiers to natural deformations over time. The anchor frame is identified as the cleanest frame capturing the object, and as time evolves, recognition becomes more difficult. We see that VITO surpasses all models on the anchor frame accuracy (+3% relative to supervised ImageNet training for *pm0*), but more importantly, the accuracy gap grows for the largest temporal displacement (+5% for *pm10*). ImageNet-A on the other hand contains ImageNet-like images that systematically fool ImageNet classifiers (i.e. 'natural adversarial examples'). On this dataset, while performance is very low across all models, VITO again shows more robustness. For additional comparison, we also evaluate two models (SIN-IN and L2-Robust ($\epsilon = 1$)) which are models trained specifically for robustness (to shape-bias and adversarial attacks respectively). While SIN-IN yields modest improvements on ImageNet-Vid-Robust, neither method approaches the gains in robustness afforded by VITO.

Finally, we evaluate robustness on the ImageNet-3DCC dataset, which contains naturalistic and synthetic corruptions applied to clean images from the ImageNet validation set [53]. To test robustness to conditions of real-world deployment, we choose the subset of corruptions designed with 3D models to be consistent with scene geometry. These include things like fog, near/far focus, motion blur, etc. and have 5 different severity levels per image. In Fig. 4.1 (Left), we plot the difference in accuracy between clean (ImageNet val) and corrupted accuracy across severity levels. This "$\Delta$-accuracy" provides a measure of how robust a model is as distortion levels increase. We see that across all corruption strengths, VITO shows increased robustness compared to supervised and self-supervised (MoCLR, DINO) ImageNet pre-trained models. The robustness gap grows significantly at the highest corruption levels, demonstrating the generality of this effect (+10% relative to supervised ImageNet training). While the robust training methods (SIN+1N1K and L2-Robust) outperform supervised and MoCLR models, VITO remains significantly more robust, demonstrating that learning from video deformations may endow a more general form of robustness than that provided by either style-transfer or adversarial images.

To quantify further the specific impact of individual components of VITO on robust recognition, we show the same plot (Fig. 4.1 (Right)), now with the ablations described in Sec 4.3. We find that all components of our method and architecture are necessary for best robustness, but in particular there is a striking split between models trained with only spatial deformations (VITO (T=0), MoCLR ImageNet, MoCLR VideoNet) and those trained with video deformations. We find that the models that learn only from image-level spatial deformations suffer significantly in robustness against all of the models that learn from video deformations.

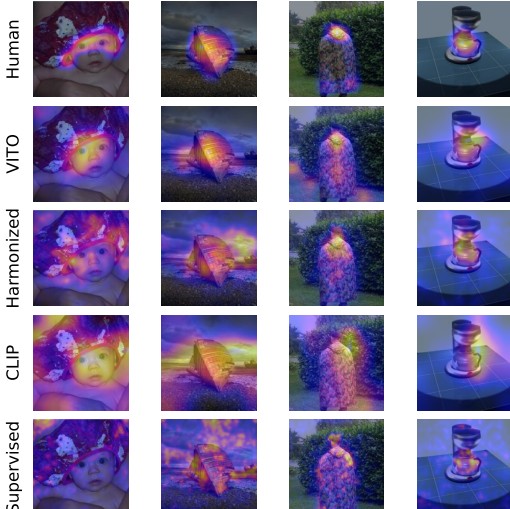

**Figure 3:** Example human saliency maps from the ClickMe dataset [71] and ResNet-50 models. Gradient-based saliency is shown for Supervised and Harmonized [72]. Attention maps are shown for CLIP and VITO model. We use multi-head attention pool weights for CLIP and average of weights from last 2 attention pooling scales in VITO.

| Method | Trained for alignment | Human Alignment |
|---|---|---|
| MoCLR [68] | ✗ | 21.4 |
| Supervised | ✗ | 34.4 |
| CLIP [58] | ✗ | 41.8 |
| Harmonized [72] | ✓ | 45.5 |
| VITO | ✗ | **47.7** |

**Table 2:** Quantitative comparison between gradient-based saliency maps (from Supervised, MoCLR, CLIP-RN50 (attention-map), and Harmonized networks), VITO attention weights, with human saliency maps using a correlation based alignment score from [71]

## 4.2 Measuring explicit human-alignment

Given that VITO representations display strong generalization across many tasks and robustness to distribution shifts, two signatures of human perceptual intelligence, we now directly ask whether they align with human perceptual representations.

**Visual saliency via contrastive attention.** We start by comparing VITO's learned attention masks to human saliency data from the ClickMe dataset [71], as well as saliency maps obtained from a collection of ResNet-50 models. For the supervised and MoCLR ResNets we use standard gradient-based saliency as in [72]. Since our model contains two attention maps at two scales of the ResNet, we upsample both maps to the image size and simply average them to obtain a single map. We compare our attention maps additionally to those obtained from the modified CLIP ResNet [58], which also utilizes attention-pooling in the final layer but is trained for image-language alignment (the canonical approach for training state-of-the-art visual language models). Because the CLIP pooling uses multi-head attention, we upsample these maps and average them across heads. Finally, we also compare to the gradient-based saliency maps from a "harmonized" model explicitly trained to align with human saliency ([72]).

Qualitatively, VITO saliency maps appear significantly more aligned with human perception maps than the supervised and CLIP ResNets (Figure 3). Surprisingly, VITO appears more aligned than the Harmonized saliency maps across the 4 examples. Quantitatively (using Spearman rank correlation) VITO outperforms the supervised, MoCLR, and CLIP models by a large margin, and even surpasses the Harmonized model which has been specifically trained for this purpose (Table 2).

This result suggests that as opposed to image-based objectives or image-language alignment, human perception of feature importance across the visual scene can be better explained as a consequence of learning what to attend to in the context of self-supervised video-based learning. We hypothesize that these attention masks could underlie the formation of high-level concepts via "semantic binding", which we investigate in Figure B.1 and Section B.1.

**Human error consistency in shape-biased tasks.** Based on this result relating to object saliency, we hypothesize that VITO may be capturing global object shape features better than traditional deep networks which have been shown to heavily rely on textural cues for classification [56].

To evaluate this quantitatively, we used a subset of the dataset proposed in [59] to test both the accuracy and consistency with human judgments of model classifications of stimuli that require

| Method | accuracy diff. ↓ | obs. consistency ↑ | ceiled error consistency ↑ |
|---|---|---|---|
| *Image pretraining* | | | |
| DINO [73] | 0.236 | 0.504 | 0.291 |
| Supervised | 0.215 | 0.511 | 0.329 |
| SIN+IN1K [56] | 0.203 | 0.527 | 0.330 |
| MoCLR [68] | 0.190 | 0.536 | 0.335 |
| L2-Robust [54] | 0.178 | 0.544 | 0.389 |
| CLIP [58] | **0.108** | **0.612** | **0.482** |
| *Video pretraining* | | | |
| R3M [74] | 0.392 | 0.359 | 0.054 |
| CycleCon [9] | 0.237 | 0.484 | 0.258 |
| VINCE [7] | 0.210 | 0.501 | 0.269 |
| VITO | **0.157** | **0.564** | **0.422** |

**Table 3:** Accuracy difference and consistency with human judgments on stimuli that are biased to requiring global-shape understanding (instead of texture) for recognition/discrimination. VITO surpasses all comparable trained models (both image and video pretraining) in all benchmarks, including those that are trained specifically to be robust (SIN+IN1K, and L2-robust). We underperform the CLIP model; however, we note that CLIP is trained with an order of magnitude more images (400M) and explicit human-language supervision.

shape-cues for effective discrimination (Table 3). Specifically, these stimuli are categorized into 4 groups: edge drawings, cue-conflict / stylized (mixing of shapes with contradictory textures through style-transfer), variable low-pass filtering (to remove high-frequency local content), uniform noise (corrupts local texture features). Based on the original methodology proposed in [5], we report the accuracy difference (from human accuracy), the raw consistency with human judgments, and ceiled error consistency (method from [5]).

We compare to supervised and MoCLR ResNets, the robust training methods cited earlier, as well as CLIP [58]. We also compare to various video pre-training methods cited earlier and another (R3M [74]), which has specifically shown to have human- and neurally-aligned representations of dynamic, object-centric scenes [75]. For all networks, we train linear classifiers on the ImageNet validation set and evaluate on the modified shape-biased stimuli. Compared with all other comparable image pretrained models, VITO achieves stronger robustness to shape-biasing transformations (lower accuracy difference relative to original images). Furthermore, VITO makes predictions more consistent with human judgements in terms of per-trial classification behavior. This is particularly surprising as VITO even outperforms the adversarially-trained robust model without requiring any explicit robust training procedure. Moreover, this improvement is not captured by prior video pretraining efforts (which are in fact far worse than the image pretraining methods). The R3M model, in particular, performs surprisingly poorly. Because the images used to collect the human judgments are modified versions of those from the ImageNet validation set, we hypothesize that this performance can be attributed to the poor transfer of the Ego4D datasets to the diverse classes present in ImageNet (contrarily to VideoNet). Indeed, the R3M model only achieves 13% accuracy on the clean ImageNet validation set (see Table B.3). Finally, we note that VITO does underperform CLIP on this benchmark; however, this comparison is not truly fair as CLIP is trained with explicit human supervision via large-scale image-language mappings. In fact, we believe that our method can be augmented with similar language supervision to improve human alignment even further.

In summary, VITO captures aspects of how humans process shape-information that cannot be captured by other strong visual models. Understanding more about this effect and what aspects of learning from videos lead to this remain interesting opportunities for future work.

## 4.3 Ablations

To understand more about how the components of VITO training contribute to its performance, we vary the different aspects of our paradigm in isolation: our method for data curation (VideoNet), multi-scale attention pooling, and details of the input data (spatial crop size and the temporal sampling scheme). We explore some ablations in detail on an example benchmark (PASCAL segmentation), but also evaluate ablations across many of the benchmarks used in this work. Finally, we provide a brief exploration demonstrating that our method scales well to larger architectures.

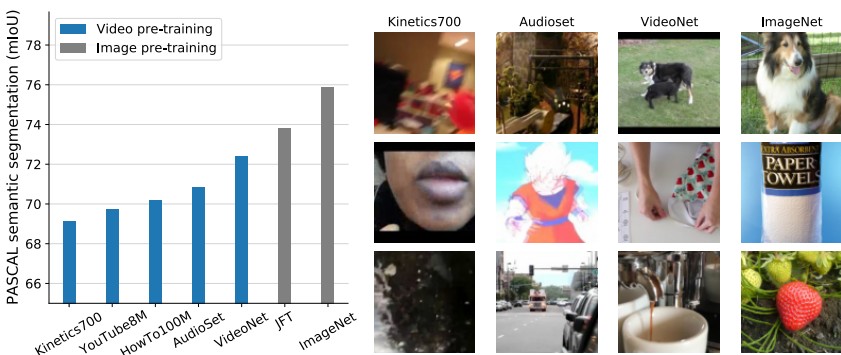

**Figure 4:** Impact of pretraining data's spatial content on representation quality. Left: transfer performance of models pretrained on single frames from image datasets (grey bars) or individual videos (blue bars). Right: example frames from different video and image datasets.

**Effect of pretraining data.** To demonstrate the effect of the pretraining data distribution on transfer performance, we pretrain a baseline MoCLR model (using 2 views) on a variety of image and video datasets, where we initially treat video datasets as collections of individual frames. We train each model for 300 ImageNet-equivalent epochs, referred to hereafter as "epochs" (i.e. 1 epoch = learning from 1.28M examples, irrespective of the dataset), such that each model benefits from the same amount of computation. Figure 4 (left) shows their transfer performance on PASCAL semantic segmentation. As expected, ImageNet pretraining works very well, but pretraining on standard video datasets results in a substantial drop in performance (e.g. $-6.8\%$ or $-5\%$ mIoU from pretraining on Kinetics700 or AudioSet). This performance gap between video and image pretraining can be attributed to a combination of increased complexity and field-of-view of video frames and domain mismatch between the dataset categories (Figure 4, right). Consistent with this, training on JFT [76], an uncurated dataset with a heavy-tailed class distribution, also results in a loss in performance. Notably, this is despite the much larger size of JFT (300M images). We find that applying the same baseline pretraining to frames from our curated video dataset performs better than existing large-scale video datasets like Audioset ($+1.6\%$ mIoU), but still underperforms image pretraining on JFT and ImageNet (Figure 4). This demonstrates the importance of aligning the distribution of video frames with that of common image datasets. We therefore use VideoNet as our primary pretraining dataset for the rest of the study. In Sec B.3 we disentangle the power of our method and dataset by confirming that each independently have strong effects: MoCLR trained on VideoNet, and VITO trained on standard datasets (Audioset or YT8M) also outperform all prior work (including models trained on much larger image datasets like JFT-300M).

**Multi-scale attention pooling.** We decompose the proposed multi-scale contrastive attention pooling to isolate the effects of multi-scale learning from those of attention pooling (Figure B.2, right). While we find only modest gains from adding attention pooling to a single-scale version of the model ($+0.2\%$ mIoU), we find that the 2-scale model (without attention pooling) improves over the single scale model more robustly ($+0.6\%$ mIoU). Interestingly, we find that the combination of the 2-scale model with attention pooling has a synergistic effect ($+1\%$ mIoU over the single-scale attention model), highlighting the importance of handling the variability in scales present in natural videos.

**Spatial and temporal augmentation parameters.** We first validate in Figure B.2 (left) our hypothesis that increasing the minimum crop-scale in the random-resized crop operation during training leads to models that generalize better to fine-grained tasks like semantic segmentation. Specifically, we find that a minimum crop scale of 0.4 (as opposed to the traditional 0.08) results in the best transfer performance ($+1.7\%$ mIoU). Note that this conclusion differs slightly from that of [37] who find more aggressive cropping to be beneficial for action recognition.

Next, to study the effect of different temporal sampling schemes, for each training example, we sample 3 views using marginal sampling of each frame from the video clip of length $T = 2.56$ seconds. This length determines the distribution of time differences between any pair of frames, and thus the time-scale over which the contrastive model learns invariances. We verify our choice by varying the total length of clips. While going to longer time-scales $T = 3.2s$ does not hurt performance much, we find a significant improvement over using shorter clips (e.g. $T = 1.28s$,

| Pretraining | Dataset | PASCAL (mIoU) | UCF101 (top-1) | IN-A (top-1) | IN-Vid (pm0/pm10) | Human error consistency |
|---|---|---|---|---|---|---|
| MoCLR | VideoNet | 72.8 | 83.0 | 2.3 | 55.5/40.5 | 0.224 |
| VITO  1scale (w/o attn) | VideoNet | 75.2 | 85.5 | 3.9 | 67.3/55.5 | 0.359 |
| VITO  1scale (attn) | VideoNet | 75.4 | 85.7 | 3.5 | 65.6/52.9 | 0.368 |
| VITO  2scale (w/o attn) | VideoNet | 75.8 | 86.2 | 4.2 | 67.4/54.9 | 0.390 |
| VITO  (T=0) | VideoNet | 74.8 | 83.2 | 3.9 | 63.9/49.5 | 0.323 |
| VITO | AudioSet | 73.8 | 84.8 | 3.4 | 55.7/42.4 | 0.401 |
| VITO | VideoNet | **76.3** | **87.4** | **5.4** | **70.6/57.2** | **0.422** |

**Table 4:** Summary of ablation models on key evaluations covering image understanding, video understanding, and human alignment on ood object recognition. In summary, it is clear that all components (pretraining data, temporal deformations, and the multi-scale attention pooling) are required for best performance across all tasks.

$+1.0\%$ mIoU; Figure B.2, center). This suggests that invariance to the rich temporal deformations present in video clips is indeed a beneficial criterion for learning fine-grained spatial representations.

**Comprehensive ablation summary.** In Table 4, we extend the above ablation studies to a more comprehensive benchmark set. In addition to the PASCAL segmentation task, we evaluate the key ablated models on video understanding (UCF101), OOD recognition (IN-A/IN-Vid) and human alignment on the shape-bias tasks specified in Sec 4.2. We confirm that all of the major methodological components (VideoNet dataset, multi-scale attention pooling, and using temporal deformations) work in concert, and are required for best performance across all tasks. Notably, we see a particularly striking dichotomy between models trained with and without temporal deformations on human error-consistency. Specifically, models trained without temporal deformations (MoCLR and VITO (T=0)) have a significant drop in human error-consistency relative to all other models trained with temporal deformations, highlighting the importance of learning these kinds of invariances.

**Scaling model architectures.** We briefly demonstrate that VITO scales to more recent larger architectures. Specifically, we show preliminary results that VITO achieves highly competitive performance on four scene understanding benchmarks using the Swin-S transformer architecture [77]. In Sec. B.4, we show that performance improves dramatically over the ResNet-50 architecture and is competitive with a strong, specialized ImageNet pretrained baseline for fine-grained scene understanding (DetCon [65]).

## 5   Discussion

**Summary.** We propose VITO, a simple method for distilling videos into visual representations. The key features of our method include improved dataset curation, adapting augmentation pipelines to appropriately handle video frames, and using attention-guided contrastive learning. With these components, VITO surpasses both prior video pretraining in spatial understanding, and image pretraining on temporal understanding and robustness. In addition to these hallmarks of human perception, VITO explicitly aligns with aspects of human saliency and image recognition behavior that are not captured by other high-performance representation learning techniques. In sum, despite the many successes in video representation learning, our results suggest that there is a great untapped potential in video pretraining as a paradigm for learning general, human-aligned visual representations.

**Limitations and Future Work.** We believe this work can be a foundation for future video pretraining efforts, as our approach is powerful, yet simple and extensible. However, we recognize that this demonstration is mostly limited to a single contrastive learning framework and ResNet-50 architecture. We leave for future work, the validation and exploration of similar analyses with larger models and other self-supervised training objectives (such as MAEs and self-distillations methods like DINO). Additionally, while we have shown the benefits of a surprisingly simple attention module for learning correspondences in video data, there are more powerful attentional architectures we can leverage along with scaling dataset size as in [10]. We have started these experiments with our exploration of Swin transformer architectures.

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

# A  Appendix: Implementation details

## A.1  Self-supervised learning

**Data pre-processing.** Each frame is randomly augmented by composing the following operations, each applied with a given probability:

1. random cropping: a random patch of the image is selected, whose area is uniformly sampled in $[s \cdot \mathcal{A}, \mathcal{A}]$, where $\mathcal{A}$ is the area of the original image, and whose aspect ratio is logarithmically sampled in $[3/4, 4/3]$. $s$ is a scale hyper-parameter set to $0.08$ when learning from ImageNet, and $0.4$ when learning from videos. Regardless, the patch is then resized to $224 \times 224$ pixels using bicubic interpolation;

2. horizontal flipping;

3. color jittering: the brightness, contrast, saturation and hue are shifted by a uniformly distributed offset;

4. color dropping: the RGB image is replaced by its grey-scale values;

5. gaussian blurring with a $23 \times 23$ square kernel and a standard deviation uniformly sampled from $[0.1, 2.0]$;

6. solarization: a point-wise color transformation $x \mapsto x \cdot \mathbb{1}_{x < 0.5} + (1 - x) \cdot \mathbb{1}_{x \geq 0.5}$ with pixels $x$ in $[0, 1]$.

The augmented frames $\boldsymbol{v}^1$ and $\boldsymbol{v}^2$ result from augmentations sampled from distributions $\mathcal{A}_1$ and $\mathcal{A}_2$ respectively. These distributions apply the primitives described above with different probabilities, and different magnitudes. The following table specifies these parameters for the BYOL framework [61], which we adopt without modification. When learning from three views, we use the distribution $\mathcal{A}_1$ to generate the third view.

| Parameter | $\mathcal{A}_1$ | $\mathcal{A}_2$ |
|---|---|---|
| Random crop probability | 1.0 | |
| Flip probability | 0.5 | |
| Color jittering probability | 0.8 | |
| Color dropping probability | 0.2 | |
| Brightness adjustment max | 0.4 | |
| Contrast adjustment max | 0.4 | |
| Saturation adjustment max | 0.2 | |
| Hue adjustment max | 0.1 | |
| Gaussian blurring probability | 1.0 | 0.1 |
| Solarization probability | 0.0 | 0.2 |

**Optimization.** We pretrain ResNet-50 using the LARS optimizer [78] with a batch size of 4096 split across 128 Cloud TPU v3 workers. We adopt the optimization details of BYOL, scaling the learning rate linearly with the batch size and decaying it according to a cosine schedule. The base learning rate is $0.3$ and the weight decay is $10^{-6}$.

## A.2  Transfer to PASCAL and ADE20K semantic segmentation

**Architecture.** We evaluate ResNet models by attaching a fully-convolutional network (FCN, Long et al. [79]) and fine-tuning end-to-end, following He et al. [16]. When evaluating Swin transformers we instead use the UperNet segmentation architecture [80].

**Data pre-processing.** During training, images are randomly flipped and scaled by a factor in $[0.5, 2.0]$. Training and testing are performed with $512 \times 512$-resolution images. When fine-tuning on ADE20K, we aditionally use photometric transformations from the mmseg[†] codebase.

---

[†]https://github.com/open-mmlab/mmsegmentation

**Optimization.** We fine-tune for 45 epochs on the PASCAL `train_aug2012` set or 60 epochs on the ADE20K `train` set. We use stochastic gradient descent with a batch size of 16 and weight decay of 0.005. The learning rate is initially set to 0.04 and decayed exponentially with a factor of $0.9^n$ where n is the iteration number. When fine-tuning external models, we sweep over the base learning rate and weight decay and report their performance given the optimal configuration. In all cases we report mIoU on the `val` set averaged across 5 runs.

## A.3 Transfer to COCO and LVIS object detection

**Architecture.** We evaluate both ResNet and Swin transformers using the FCOS$^\star$ architecture, following Hénaff et al. [81]. FCOS$^\star$ is the implementation of a single-stage detector based on FCOS [82], and improved with the collection of techniques from Wu et al. [83], Zhang et al. [84], and Feng et al. [85], full details can be found in Hénaff et al. [81].

**Data pre-processing.** The target resolution is 800×1024. During testing, an image is resized by a factor $s$ while preserving the aspect ratio, such that it is tightly contained inside the target resolution, and then padded. When fine-tuning, the image is rescaled by a factor of $u \cdot s$ where $u$ is uniformly sampled in $[0.8, 1.25]$, and is then cropped or padded to the target resolution.

**Optimization** The network is fine-tuned for 30 epochs on the COCO `train2017` set or the LVIS `v1_train` set. We use AdamW [86] with weight decay $10^{-4}$, base learning rate of $10^{-3}$, and batch size 128 split across 16 workers. The learning rate rises linearly for $\frac{1}{4}$ of an epoch, and is dropped twice by a factor of 10, after $\frac{2}{3}$ and $\frac{8}{9}$ of the total training time. We report mAP on the COCO `val2017` set and the LVIS `v1_val` set, averaged across 5 runs.

## A.4 Transfer to DAVIS video segmentation

As a further test of scene understanding, we assess whether learned representations can continue to recognize parts of an object as they evolve over time. Video object segmentation, specifically in its semi-supervised setting, captures this ability, which we evaluate on the DAVIS'17 benchmark. Having evaluated a learned representation on a video independently across frames, we segment these features with nearest neighbor matching from frame to frame, given a segmentation of the first frame. In this way, the segmentation is propagated according to the similarity of the representation across space and time. We reuse the segmentation procedure from Xu and Wang [8] without modification, and report region ($\mathcal{J}$) and boundary quality ($\mathcal{F}$).

## A.5 Transfer to UCF-101 action recognition

We evaluate action recognition classification on the UCF101 dataset [3]. We follow the procedure for finetuning used in [9] which is based on [87]. We utilize clips of 2 seconds in length at 12fps. Each frame is processed by the ResNet-50 backbone. Clip representations are obtained by one of three methods for temporal integration:

1. Average pooling is the standard baseline, producing a 2048-d vector output for a clip which is then fed to and one fully connected (2048×101) layer for predicting the action class.

2. MS avg-pool: we pool the block3 representations (1024-d) over the two subclips of 1s each that make up the larger clip. This is done because the features at this scale have smaller receptive fields and are selective for less complex content. Then we concatenate the two (1024-d) vectors with the average pooled feature from the block4 output to get a single 4096-d vector for each clip that again is fed through a fully-connected layer to predict the action class. By concatenating the two subclip representations, the fully-connected layer can in fact compute complex temporal relationships such as differences etc. along with the final layer's invariant representation that is pooled for the full clip.

3. MS temp-attn: We perform the same methodology as above for integrating multiple scales, but replace the average pooling over time with an attention pooling layer. Given representations for an L-frame clip $z \in \mathbb{R}^{B \times C \times L}$ at a given scale, we compute temporal attention weights $w_t \in \mathbb{R}^L$ where $w = f(z)$. We choose

$f$ to be $\tanh(Wz)$ where $W \in \mathbb{R}^{C \times 1}$ is a linear weighting of channels. Finally the pooled representation $v = \sum_L w_t \cdot z$

We show results using method 1 in the main text and demonstrate the improvements from methods 2 and 3 in Appendix Table B.5. 10 clips are sampled from each video and the predictions of the clips are averaged for the final results. We fine-tune for 16 epochs using the ADAM optimizer with a multi-step LR decay schedule at epochs 6, 10, and 14. The initial learning rate is set to 0.0001. The implementation is adopted from `https://github.com/facebookresearch/AVID-CMA`.

## A.6 Transfer to ImageNet classification

For all models we freeze the ResNet-50 encoder (which outputs 2048-d embeddings). We then train a linear head to classify the 1000 categories in the ImageNet training set using the standard split. To train the classifier, we use the SGD optimizer with nesterov momentum and momentum parameter equal to 0.9. We use weight-decay of 0 and sweep the learning rate for each model in the range [0.4, 0.3, 0.2, 0.1, 0.05] and pick the best classifier based on ImageNet validation accuracy.

## A.7 Transfer to out-of-distribution evaluations

For all OOD evaluations, we evaluate on datasets that utilize all (or subsets) of the ImageNet validation set. Therefore, for these evaluations we use the pre-trained encoder and linear classifier (trained as in Sec A.6). We freeze the encoder and linear classification head and evaluate task performance on images from either the ImageNet-A, ImageNet-vid-robust, and Imagenet-3DCC datasets. For ImageNet-A and ImageNet-vid-robust, we use the evaluation code and method from [4].

For ImageNet-3DCC, we do not use the entire corruption set because we wanted to specifically test models under the more natural 3-d corruptions. As is described in [53], the dataset can be broken down into two sets of corruptions: 3-d informed corruptions (using a depth model to generate natural corruptions informed by 3-d information) and standard 2-d noise and artifacts (like in ImageNet-C). For our experiments, we chose to evaluate specifically on the 3-d corruptions, which were found to induce larger robustness effects for evaluating standard networks [53]. Nevertheless, we found similar results when evaluating robustness to 2-d noise and artifacts. All images from the following classes of corruptions were used for evaluation: far focus, near focus, fog, flash, xy motion blur, z motion blur, view jitter.

## A.8 Alignment with human saliency

Human saliency measurements are obtained from the ClickMe dataset. Alignment is measured as the Spearman rank correlation between model and human saliency averaged over the dataset, normalized by inter-rater alignment of humans.

## A.9 Human error consistency evaluation

We evaluate accuracy and human error consistency on shape-bias datasets using the code from `https://github.com/bethgelab/model-vs-human/tree/master`. We choose the subset of images that remove textural cues in different ways (forcing humans and models to utilize global shape during discrimination): edge drawings, cue-conflict stimuli, graded low-pass filtering, and uniform gaussian noise. We report three metrics from [59]:

1. Accuracy difference: measure of human vs model classification accuracy on each OOD dataset and then averaged.

2. Observed consistency: measures the fraction of samples for which humans and a model get the same sample either both right or both wrong.

3. Error consistency: Score that measures whether there is above-chance consistency. This is important because e.g. two decision makers with 95% accuracy each will have at least 90% observed consistency, even if their 5% errors occur on non-overlapping subsets of the test data (intuitively, they both get most images correct and thus

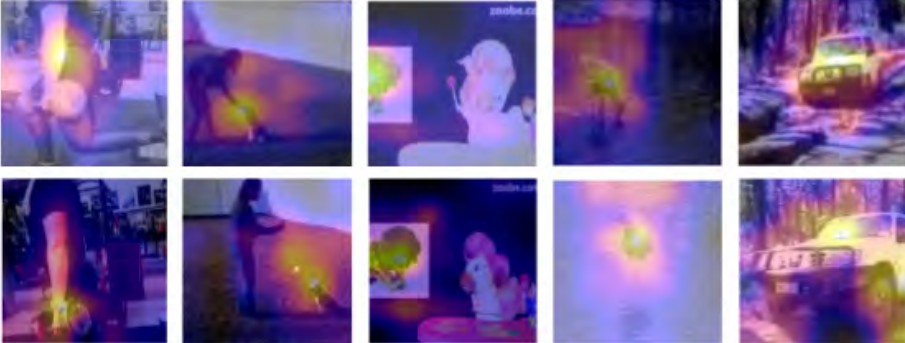

**Figure B.1:** Example augmented frames with overlaid (resized) learned attention masks. Attention is computed from the output of the final block of the VITO trained ResNet-50. Crucially, the attention masks are computed independently, such that the attention module can only use spatial cues.

observed overlap is high). Error consistency indicates whether the observed consistency is larger than what could have been expected given two independent binomial decision makers with matched accuracy [5].

The mathematical details on each of these metrics are provided in [5].

# B    Appendix: Additional results

## B.1    Semantic binding with contrastive attention pooling

The ablation study demonstrated that multi-scale attention improves the performance of VITO in semantic segmentation. To probe why this may be, we visualize and interpret the learned attention masks (Figure B.1). For simplicity, we only visualize the masks from the coarsest scale (output feature map), but the interpretation naturally extends to the multi-scale version as these masks are learned with independent attention modules.

Because the attention masks are not computed jointly across each view, for a given video frame, the attention module must marginalize over the training data to make a statistical prediction—what should be attended to in the first view in order to minimize the contrastive loss across possible second views? Specifically, the attention must focus on content that is most likely to be stable across time while still being discriminative (or unique) relative to other frames from other videos. Different examples appear to trade-off these criteria differently, yet systematically. For example, in the third column of Figure B.1 even though the animated characters on the right side of both frames may be discriminative content, the attention module has learned to focus on the static picture on the left as it is the content that is most likely to be stable across time. For this pair of frames the prediction is correct—the attention disregards content that is changing too abruptly—despite not having access to motion cues. On the other hand, the example in the fourth column demonstrates a scenario where the model has attended to stable, but primarily discriminative content (the bird) rather than the background, which is also very stable but most likely less unique relative to other videos.

Even beyond the ability to localize stable, yet discriminative content, it seems that our method also enables "semantic binding" of visually different, but semantically related features. This can be seen in the first pair of frames, as the model has learned to associate an arm or elbow (in the first frame) with the dumbbell (in the second frame), demonstrating an understanding that these two semantically related concepts co-occur and thus are predictive of one another given the right embedding.

Binding co-occuring features appears as an intuitive explanation for why these representations would perform well on semantic segmentation. It is particularly interesting that training end-to-end with a standard contrastive loss can produce complex behavior reminiscent of the DINO approach [19] even though we use a single, two-layer MLP attention module as opposed to large-scale transformer architectures which use attention throughout the network.

## B.2 Ablating the components of VITO

In Figure B.2 we demonstrate on an example scene understanding task (PASCAL) how VITO is impacted by crop-scale, clip length, and the type of attention pooling used (or not used). In Figure B.3 we additionally do a deeper analysis of the temporal sampling scheme and demonstrate that our choice performs best across tasks and is arguably the most natural.

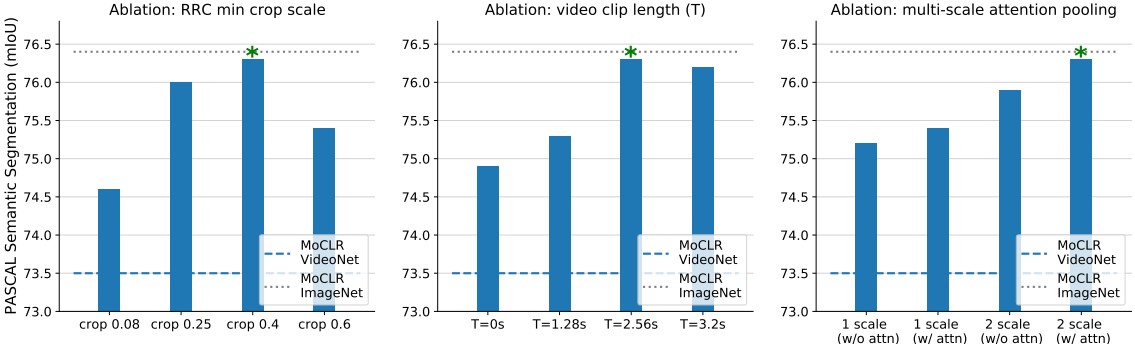

**Figure B.2:** Effects of crop scale, natural augmentations, and multi-scale attention on representation quality. All ablations are performed relative to VITO's configuration (denoted by a green asterisk) which uses 2-scale attention pooling, a less aggressive crop scale of 40%, and natural augmentations uniformly sampled in a window of length T = 2.56s. We also compare to our baseline MoCLR model trained on single frames, either from ImageNet (dotted gray line) or VideoNet (dashed blue line). All models are evaluated by transferring to PASCAL semantic segmentation.

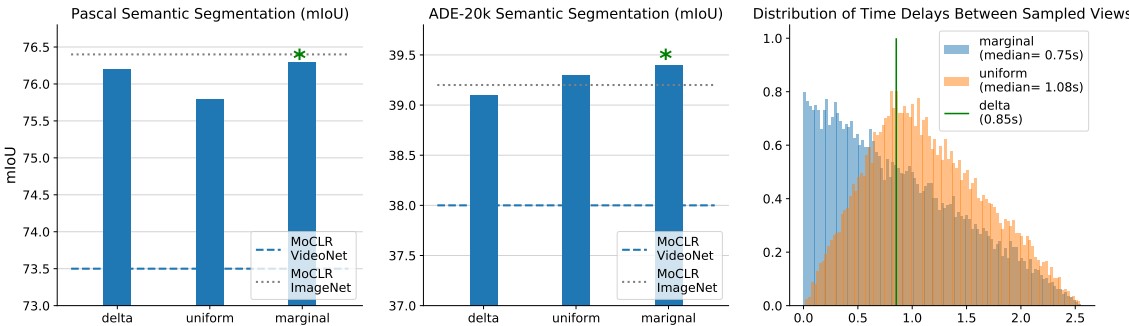

**Figure B.3:** Ablating different temporal sampling schemes. *Delta* refers to fixed time sampling beteween frames as in Gordon et al. [7]. *Uniform* refers to chunking time into non-overlapping blocks and uniformly sampling within each chunk as in Xu and Wang [8]. *Marginal* sampling (ours) refers to simple uniform sampling from the full video clip of length $T = 2.56s$. First two panels show that marginal sampling is best overall across transfer to PASCAL and ADE20K. Third panel shows the distribution of absolute time-differences between any two pairs of frames under each sampling scheme (assuming 3 views are sampled per clip). Our marginal sampling scheme is arguably the most natural as the mode of the distribution is at 0, meaning that it is not biased to over-represent any specific time difference (similarly to the random-resized crop operation in space).

## B.3 Dataset and method ablations

In Table B.1 we show that both our learning objective VITO and choice of dataset, VideoNet, are important for achieving top performance. However, these results also show that we can outpeform exisitng video pretraining even when using

standard datasets like Audioset and YT8M. In addition, by comparing to MoCLR trained on JFT-300M, we demonstrate the benefits of our method are not the result of simply having more frames of training data.

| Pretraining | Dataset | Epochs | Semantic segmentation | | Object detection | |
|---|---|---|---|---|---|---|
| | | | PASCAL | ADE20K | COCO | LVIS |
| MoCLR | VideoNet | 200 | 72.8 | 37.5 | 42.6 | 24.6 |
| VITO | YT8M | 200 | 71.8 | 37.8 | 42.7 | 24.6 |
| VITO | AudioSet | 200 | 73.6 | 38.5 | 43.2 | 25.0 |
| VITO | VideoNet | 200 | **75.5** | **39.2** | **43.6** | **25.6** |
| MoCLR | JFT-300M | 200 | 74.3 | 38.7 | 43.2 | 25.4 |

**Table B.1:** VITO dataset and method ablations. We compare the baseline method MoCLR trained on VideoNet to demonstrate the impact of our methodology. VITO on VideoNet performs significantly better due to the methodological improvements (attention pooling, adaptation of spatial and temporal augmentations). We also evaluate VITO on traiditonal video datasets such as YT8M and AudioSet. We note that these numbers still greatly outperform prior video pretraining (See Table B.3. However the imapct of the VideoNet dataset is clear as the best model is VITO trained on VideoNet. Finally, we show that VideoNet *does not* simply provide benefits due to increased number of total frames vs. ImageNet. In fact, MoCLR trained on JFT-300M has an order of magnitude more frames and yet still underforms.

## B.4 Scaling architectures

Here we demonstrate that VITO scales effectively to more powerful Swin transformer architectures. Results on scene understanding benchmarks improve greatly over ResNet-50 models and are competitive with specialized fine-grained scene understanding models from recent literature [65]. See Table. B.2

| Pretraining | Dataset | Backbone | Semantic segmentation | | Object detection | |
|---|---|---|---|---|---|---|
| | | | PASCAL | ADE20K | COCO | LVIS |
| VITO | VideoNet | R50 | 76.3 | 39.4 | 44.0 | 25.7 |
| MoCLR | VideoNet | Swin-S | 78.6 | 43.7 | 48.4 | 32.7 |
| VITO | VideoNet | Swin-S | **81.3** | **46.1** | 49.8 | **33.5** |
| Detcon$_B$ | ImageNet | Swin-S | **81.4** | **46.1** | **50.4** | 33.1 |

**Table B.2:** VITO scales to larger model architectures (Swin-S), improving performance compared to the ResNet-50 baseline and remaining competitive with a strong ImageNet pretrained baseline (Detcon) from Hénaff et al. [65].

## B.5  Comparisons on additional scene understanding tasks

VITO outperforms all prior video pretraining (of image representations) on scene understanding tasks. In addition to the evaluations in the main text, we add PASCAL segmentation, LVIS object detection, ImageNet-1K classification. VITO remains highly competitive with the best ImageNet pretraining on these tasks. (See Table B.3).

| Video Pretraining | Dataset | Semantic segmentation | | Object detection | | Classification |
| | | PASCAL | ADE20K | COCO | LVIS | IN-1K |
|---|---|---|---|---|---|---|
| Random Init | | 53.0 | 27.9 | 39.0 | 21.1 | - |
| *Methods pretraining on video datasets* | | | | | | |
| R3M [74] | - | - | - | - | - | 13.3 |
| VFS [8] | K400 | 63.9 | 31.4 | 41.6 | 23.2 | - |
| VIVI [69] | YT8M | 65.8 | 34.2 | 41.3 | 23.2 | 62.6 |
| VINCE [7] | R2V2 | 69.0 | 35.7 | 42.4 | 24.4 | 54.4 |
| CycleContrast [9] | R2V2 | 69.2 | 35.6 | 42.8 | 24.5 | 55.6 |
| MMV TSM [70] | AS + HT | 70.6 | 32.5 | 41.3 | 24.2 | 51.4 |
| VITO | VideoNet | **76.3** | **39.4** | **44.0** | **25.7** | **66.2** |
| *Methods pretraining on ImageNet* | | | | | | |
| Supervised | ImageNet | 71.3 | 33.5 | 44.2 | 25.2 | 76.1 |
| BYOL [61] | ImageNet | 76.1 | 38.8 | 43.7 | 25.5 | - |
| MoCLR [68] | ImageNet | 76.4 | 39.2 | 43.9 | 25.8 | 71.4 |
| DINO [19] | ImageNet | 76.1 | 39.0 | 44.3 | 26.4 | 75.3 |

**Table B.3:** Image and pretraining evaluated on object-detection, semantic segmentation, and ImageNet-1K classification.

## B.6  Comparison to image pretraining on video-based tasks

Here we demonstrate more thoroughly that compared with image pretraining methods (image backbones), we perform significantly better on video-level tasks. On both DAVIS segmentation (Table B.4) and UCF-101 action recognition (Table B.5), VITO outperforms strong ImageNet trained baselines and methods pretrained on video datasets.

| Pretraining | Dataset | $\mathcal{J}_m$ | $\mathcal{F}_m$ |
|---|---|---|---|
| *ImageNet pretraining* | | | |
| Supervised | ImageNet | 63.7 | 68.4 |
| MoCo [16] | ImageNet | 63.2 | 67.6 |
| DetCon$_B$ [65] | ImageNet | 63.1 | 66.4 |
| MoCLR [68] | ImageNet | 63.1 | 67.8 |
| BYOL [61] | ImageNet | 63.8 | 69.4 |
| *Video pretraining* | | | |
| VINCE [7] | Kinetics | 63.4 | 67.8 |
| TimeCycle [88] | VLOG | 41.9 | 39.4 |
| UVC [89] | Kinetics | 54.5 | 58.1 |
| CRW [44] | K400 | 64.8 | 70.2 |
| VFS [8] | K400 | 65.3 | 70.2 |
| VITO | VideoNet | **65.5** | **70.8** |

**Table B.4:** VITO significantly outperforms all image-pretraining baselines on DAVIS 2017 video segmentation. VITO also outperforms many recent successful video pretraining methods.

| Pretraining | Dataset | Backbone | Top-1 |
|---|---|---|---|
| *Video architectures* | | | |
| Supervised [90] | ImageNet | I3D | 67.1 |
| VideoMoCo [91] | K400 | R(2+1)D | 78.7 |
| Temporal-ssl [92] | K400 | R(2+1)D | 81.6 |
| VTHCL [93] | K400 | 3D-R50 | 82.1 |
| CoCLR [94] | K400 | S3D | 87.9 |
| CVRL [34] | K400 | 3D-R50 | 92.9 |
| $\rho$-BYOL [37] | K400 | 3D-R50 | 95.5 |
| Supervised [95] | K400 | I3D | 95.1 |
| *Image architectures* | | | |
| OPN [96] | UCF101 | VGG-M | 59.8 |
| TCE [97] | K400 | R50 | 71.2 |
| CycleContrast [9] | R2V2 | R50 | 82.1 |
| MoCLR [68] | ImageNet | R50 | 85.5 |
| BYOL [61] | ImageNet | R50 | 85.6 |
| VITO  (avgpool) | VideoNet | R50 | **87.4** |
| VITO  (MS-avgpool) | VideoNet | R50 | **88.5** |
| VITO  (MS-attnpool) | VideoNet | R50 | **89.4** |

**Table B.5:** VITO outperforms all image representations when finetuning for UCF101 action recognition, using temporally-pooled frame-level representations. VITO's performance is even competitive with many video architectures.

