# OpenReview forum: "Self-supervised video pretraining yields robust and more human-aligned visual representations"
_NeurIPS.cc/2023/Conference — NeurIPS 2023 poster_

### Official Review · Reviewer_jN1F · 2023-06-26

**Soundness:** 3 good
**Presentation:** 3 good
**Contribution:** 3 good
**Rating:** 6
**Confidence:** 3

**Summary:**

The research question addressed by the paper is whether self-supervised training on
videos leads to image features that are better aligned with human perception. To that
end, the authors propose a contrastive learning method that leverages natural
changes over time as different views, beyond adapted image augmentations. Moreover, an
attention mechanism is introduced that allows the model to compare image pairs based
on features from selected regions only. The model is trained on videos that have been
curated to better match the ImageNet category distribution than existing video datasets.
Trained this way, the model performs better than previous methods when transferred
to several standard benchmarks and shows a better alignment with human vision.

**Strengths:**

- Training computer vision models to better align with human perception is an active
  area of research with the potential to improve, e.g., the robustness of current
  approaches. Self-supervised training on natural videos is closer to how humans learn
  than standard training methods. Exploring this direction therefore is well motivated.
- The authors compare to a broad range of previous models on several computer viion
  benchmarks. The newly proposed method improves over previous approaches in most cases.
- Several ablation studies are performed and reported in the supplement to disect the
  influences of the contributions on the improved performance.


**Weaknesses:**

The paper aims at better aligning neural network features with human perception.
However, the range of models tested for alignment with human perception is much smaller
than the range of models tested for performance on computer vision bechmarks. In
particular, none of the other video based self-supervised methods is tested and none
of the ablation studies considers alignment with human vision.

The title of the paper, "Self-supervised video pretraining yields human-aligned visual
representations", and the introduction therefore raised wrong expectations for me. Due to the
missing ablations, one cannot judge whether the improved human alignment is due to the
video pretraining or other contributions, such as the new training set or the
multi-scale contrastive attention pooling. Moreover, I find the title too bold given
that other methods perform much better in terms of human error consistency according to
the official benchmark repository (e.g., CLIP).

**Update:** Both points raised above have been addressed during the rebuttal. Therefore I improved my rating of this paper.

**Questions:**

As described above, the paper has a much stronger focus on performance in terms of
computer vision (where the proposed method is very competitive) than alignment with
human vision. I would recommend to change the title and introduction to better align
with this focus and I am happy to increase my score if this issue is addressed.

L186ff needs clarification for me: When evaluating video understanding, the paper claims
that "VITO learns features that capture finegrained temporal deformations of objects".
However, VITO is based on image networks so that spatio-temporal features cannot be
learned.

**Limitations:**

The authors discuss limitations of the paper due to the focus on a single architecture
(ResNet-50). Potential societal impacts are not discussed.

---

> ### Author Rebuttal · Authors · 2023-08-10
>
> We thank the reviewer for the comments and address the concerns below.
>
> **“none of the other video based self-supervised methods is tested and none of the ablation studies considers alignment with human vision.”**
>
> We agree this was a limitation of our current results and have addressed this by greatly expanding the models evaluated on our benchmarks and human alignment. (See global response). The results stand that VITO outperforms all prior video pretraining on both the visual tasks and human alignment, and we have verified that these effects are strongest only with the combination of the contrastive attention pooling and the use of temporal deformations during training.
>
> **“Due to the missing ablations, one cannot judge whether the improved human alignment is due to the video pretraining or other contributions, such as the new training set or the multi-scale contrastive attention pooling.”**
>
> We refer the reviewer to our general response and ablation figures in the rebuttal PDF for a complete analysis of the impact of the ablations on the improved human alignment and robustness. In summary, we have verified that all components (dataset, contrastive attention pooling, and temporal augmentations) are necessary for the strongest performance. In addition, we find a particularly strong relationship between robustness/good human alignment with models that have been trained with the temporal augmentations.
>
> **“Moreover, I find the title too bold given that other methods perform much better in terms of human error consistency according to the official benchmark repository (e.g., CLIP).would recommend to change the title and introduction to better align with this focus and I am happy to increase my score if this issue is addressed.”**
>
> We refer the reviewer to the global response clarifying the intentions behind our claims and title (which we will incorporate in the text), and are happy to discuss/change based on what the reviewer thinks would make the most appropriate title. Regarding comparisons with CLIP, we first would like to note that CLIP in fact does not outperform our method on the saliency benchmarks which are a strong test of one aspect of perceptual alignment. Additionally regarding the shape-bias alignment, we have now included CLIP in our evaluations and while we find that it performs better than VITO, it is not by a very large margin. Moreover, CLIP is also trained with large-scale language supervision (400M image-text pairs), which is far larger than VideoNet, so we find it significant that we can obtain similar gains over standard ImageNet pretraining with far less data and supervision than CLIP training. More relevant are comparisons to image- and video-only self-supervised learning methods, where VITO outperforms prior work. Nevertheless, it would be very interesting to combine our proposed self-supervised video pretraining with multimodal language supervision.
>
> **“When evaluating video understanding, the paper claims that "VITO learns features that capture finegrained temporal deformations of objects". However, VITO is based on image networks so that spatio-temporal features cannot be learned.”**
>
> We fully agree that VITO cannot capture spatio-temporal features explicitly. However, as seen in the ablations (see main response), VITO does in fact perform significantly better when trained with temporal deformations. We believe this is due to the fact that VITO learns to attend to the features in an image which are more likely to be predictable under the fine-grained temporal deformations of objects. These features can be quite different than those that are predictable under standard image-based augmentations. Therefore, even though VITO does not capture the deformation explicitly, it is learning to become robust to the fine-grained temporal deformations of objects. We will clarify this in the main text.

---

> > ### Comment · Reviewer_jN1F · 2023-08-10
> >
> > Thank you very much for your detailed response. I appreciate the extension of the results regarding the alignment with human vision, now including both an ablation study and more extensive comparisons to prior work. In my view these results strengthen the focus of the paper and I will improve my rating accordingly.
> >
> > Thank you for clarifying the focus of your paper. As written in my original review, I see training on videos is a well motivated direction regarding alignment with human vision. In the general response you clarify that you "do not mean to claim we have learned the most human-aligned, general visual representation". However to me the title seems to exactly claim that, a more appropriate alternative could be "Self-supervised video pretraining improves human-alignment of visual representations".

---

> > > ### Author Response · Authors · 2023-08-10
> > >
> > > We thank the reviewer very much for their prompt response and for appreciating our new results and being willing to update their score.
> > >
> > > Regarding the title, yes we see the reviewer's concern and are happy to change the title to the reviewer's suggestion as we agree that it is more appropriate. We hope that this alleviates any remaining concerns.

---

> > > > ### Author Response · Authors · 2023-08-15
> > > > **Title clarification**
> > > >
> > > > On further review, we believe that some of the confusion regarding the claims of our title were a result of overloading the term “human-alignment”. As a result, we propose a more explicit change to our title: “Self-supervised video pretraining yields general, robust, and more human-aligned visual representations”.
> > > >
> > > > Coupled with the other rebuttal responses, we hope that this better conveys the focus and intentions of our paper. Given these changes, we would like to thank the reviewer for their suggestions and confirm that they will be updating their rating.

---

> > > > > ### Comment · Reviewer_jN1F · 2023-08-18
> > > > >
> > > > > Thank you for your response. I believe the proposed title is clearer than the original title and helps to clarify the focus of the paper, alongside incorporating your discussion from the global response.
> > > > >
> > > > > I updated my rating in the official review above.

---

### Official Review · Reviewer_3hVk · 2023-07-03

**Soundness:** 3 good
**Presentation:** 4 excellent
**Contribution:** 3 good
**Rating:** 7
**Confidence:** 4

**Summary:**

This paper introduces a new video dataset (VideoNet), and architectual adjustments (VITO) that enables allows pretraining on video datasets to improve performance on downstream transfer tasks, such as segmentation, object detection, and generalization. The VITO network uses a ResNet backbone, and architectural changes include extracting spatial feature maps at the two penultimate blocks, applying a learned softmax function to obtain a weighting over features, and then projecting this into a final feature map. This feature extractor is trained using the InfoNCE loss on two sampled video frames, with additional image-space augmentations applied. Additional experiments in the supplementary material investigate the use of transformer backbones. The VideoNet dataset retrieves video clips matching the ImageNet categories, and applies an image classifier on these videos to filter for the correct category. Additional experiments compare the attention maps from the VITO model and other baselines to human ground truth from the ClickMe dataset.

**Strengths:**

- The proposed method and training dataset presents strong empirical results. It outperforms, or is competitive with, image pretraining on ImageNet, video pretraining, or robust image pretraining on a variety of transfer tasks. Notably, it demonstrates improved robustness on datasets with image corruptions even when compared to models trained for robust classification.
- Ablations evaluate the independent contributions of both the dataset on the model design. With the same VITO model, but different datasets, VideoNet can outperform AudioSet as a prior video dataset on segmentation and object detection tasks. On the same dataset (VideoNet) but different models (MoCLR vs VITO), VITO also outperforms the prior method. (Table B.1)
- On additional experiments on Swin transformers show promising improvements. I think additional details on the transformer setup, and  which spatial feature maps are extracted in the transformer would be helpful.
- This paper is well written and clear to follow.


**Weaknesses:**

- The benefit of VITO seems to be primarily on transfer tasks. On standard classification, it falls short of ImageNet pretraining by ~10%, but it is still outperforming other video tasks. It is also slightly lower on object detection compared to the DINO representation. However, I think there are still valuable insights to be gained from the methodology and dataset presented here.
- There are some additional parts that could benefit from clarification. Please see the below questions section.


**Questions:**

- L124: what is the softmax normalized over?
- L130: is $a_\xi$ also a target network? It may be helpful to also illustrate the network $g$ in Fig. 1
- Fig 3: Why was the CLIP Resnet chosen for the attention map, as opposed to the attention maps extracted from CLIP or DINO transformer models?

**Limitations:**

Limitation are adequately addressed. The current setup explores primarily the ResNet backbone with the InfoNCE loss, but there are promising initial results with transformer models.

---

> ### Author Rebuttal · Authors · 2023-08-10
>
> We thank the reviewer for their positive feedback and now address the weaknesses/questions.
>
> **Weakness 1: lower ImageNet validation numbers**
>
> Yes VITO does underperform supervised ImageNet pretraining on ImageNet classification by ~10% and other SSL ImageNet pretraining methods by 5-7%. This however can be attributed to the fact that ImageNet classification is an “in-distribution” task for models pretrained on ImageNet. Even though a separate validation set is used, the training images are sampled from the same distribution, as they share properties such as single-object field-of-view, centrally cropped, high resolution etc. Even though VideoNet matches the overall class distribution to ImageNet, we cannot expect our models to transfer as well to ImageNet as it is still out-of-distribution in many respects. Indeed VideoNet frames contain multiple objects and more global scenes, strong motion deformations, and wider variability in resolution and quality. Even still, VITO outperforms all ImageNet trained models on benchmarks based on ImageNet distribution shifts (ImageNet categories under distribution shifts). For example, VITO strongly outperforms all ImageNet pretraining on ImageNet-Vid and ImageNet-3DCC. These recognition benchmarks are arguably better tests of generalization under real-world conditions.
>
> Additionally, while VITO is competitive on all other scene understanding benchmarks and strongly surpasses ImageNet pre-training on transfer to video-based tasks (DAVIS, UCF). Therefore, on average, VITO generalizes across all tasks significantly better than the comparable ImageNet pretrained models.
>
> **Question 1: L124: what is the softmax normalized over?**
>
> The softmax is normalized over space such that the attention-weights across space to sum to 1. This encourages competition across spatial locations, forcing the attention to be more localized.
>
> **Question 2: L130: is a_xi also a target network? It may be helpful to also illustrate the network g in Fig. 1**
>
> A_xi is also a target network as the EMA is applied to all parameters for the target.  We understand that missing g in Fig 1 can be confusing and will update the figure and caption to emphasize it’s role.
>
> **Question 3: Fig 3: Why was the CLIP Resnet chosen for the attention map, as opposed to the attention maps extracted from CLIP or DINO transformer models?**
>
> We choose the CLIP ResNet for attention map comparison as it provides an architecturally matched comparison to ours. We fully agree that scale in data and architecture can produce improvements, but our goal was to understand the impact of video pretraining on the image representations for this we require comparisons that are matched in model capacity to isolate these effects.

---

> > ### Comment · Reviewer_3hVk · 2023-08-11
> > **Response to author rebuttal**
> >
> > Thanks to the authors for the response and the additional clarifications and ablation experiments. I agree with reviewer jN1F that perhaps human alignment was not the principle focus of this paper, but overall I find the methodology presented here to demonstrate compelling results across a variety of tasks. I will retain my original rating.

---

### Official Review · Reviewer_Xjw1 · 2023-07-07

**Soundness:** 3 good
**Presentation:** 3 good
**Contribution:** 2 fair
**Rating:** 6
**Confidence:** 4

**Summary:**

The paper proposes a self-supervised method for learning image representations from videos.
Towards this end, a procedure to curate video datasets most suitable for such pre-training is proposed by selecting videos that best match the distribution of visual classes found in ImageNet.
Secondly, this dataset is leveraged through a contrastive training approach where the common global average pooling of video frame features is replaced with an attentional module that learns to select spatial regions of several network layers to construct the image embedding.
The learned representations are evaluated on a large set of downstream tasks, including spatial scene understanding (segmentation + detection), video understanding (segmentation + action recognition), and robust image classification.

**Strengths:**

- The method is very well presented, and the paper is well written
- The downstream evaluation of the learned image representation is extensive, and the model achieves good performance across a variety of tasks
- The finding that a video dataset with similar category distribution to ImageNet appears to perform considerably better than many existing video datasets is interesting (see Fig 4)

**Weaknesses:**

- I would have preferred to see more extensive ablation experiments to support the technical contributions of the paper (e.g., the attentional pooling module for contrastive learning). These ablations are currently only reported on a single task and dataset. It would be more convincing to see the influence on all the downstream tasks in Table 1 and to summarize these ablation results in a Table. As it is, it is unclear if the benefits of the proposed training are consistent across tasks or if most of the benefit is due to the data curation process.
- The data curation process relies fundamentally on reproducing the object category distributions of ImageNet in a video dataset. While it is interesting that this appears to lead to a benefit across many downstream tasks, it is unclear what explains these benefits. Is it greater visual diversity, lesser class imbalance, or maybe some other property?

**Questions:**

I would appreciate it if the authors could address the weaknesses outlined above. I'm particularly interested in how the technical contributions affect downstream results across multiple benchmarks (i.e., how do the ablations look when evaluated across the multiple benchmarks in Table 1)?

**Limitations:**

Limitations are adequately addressed.

---

> ### Author Rebuttal · Authors · 2023-08-10
>
> We thank the reviewer for the comments and address the concerns below.
>
> **Weakness 1: More extensive ablations**:
>
> See our global response for a detailed discussion of more extensive ablations we have now performed. Notably, we have verified that all components (dataset, contrastive attention pooling, and temporal augmentations) all are necessary for the strongest performance. In addition, we find a particularly strong relationship between robustness/good human alignment with models that have been trained with temporal augmentations. We believe that strengthening this link is important and will update our main text accordingly.
>
> **Weakness 2: unclear what explains VideoNet performance**
>
> We agree with the reviewer that there is still much more to unpack about the VideoNet curation and what aspects are necessary or not. We have provided ablations comparing to YT8M and training on the far more diverse JFT-300M image dataset in Supp Table B.1. In both cases, VideoNet outperforms by a large margin across the scene understanding benchmarks. This suggests that the key feature has to do with the shape of the class distribution rather than just greater visual diversity. Specifically, we hypothesize that having densely sampled classes and lesser class imbalance (removing long tails) greatly improves performance because it provides more difficult yet useful negative examples for the contrastive loss. It remains to be seen whether we can test this more empirically by controlling for different properties of the distribution and evaluating, but we believe our work will be a solid basis for this investigation.

---

> > ### Comment · Reviewer_Xjw1 · 2023-08-20
> >
> > I read the rebuttal and the other reviews. I appreciate the novel results in the rebuttal and agree with the overall positive assessment of the other reviews. I vote for accepting this paper.

---

### Official Review · Reviewer_Pto4 · 2023-07-10

**Soundness:** 3 good
**Presentation:** 3 good
**Contribution:** 3 good
**Rating:** 5
**Confidence:** 4

**Summary:**

This paper studies how to take advantage of natural temporal distortions in video to learn image spatial representations. The paper first proposes a VideoNet dataset that filters the video data from common video datasets by an ImageNet classifier. The paper further proposes to multi-scale attention pooling to improve the baseline algorithm. The algorithm trained on the proposed the dataset is evaluated on both image and video datasets. The paper also studies the human alignment of the learned attention.

**Strengths:**

1. Pre-training image representations with video datasets is a natural and valuable idea to explore.

2. The VideoNet dataset could be a valuable contribution to the community provided the authors have a plan to release it publicly.

3. The paper provided abundant experiments to demonstrate the effectiveness of the proposed algorithm.

**Weaknesses:**

1. Although using an ImageNet classifier to filter the video data is a practical idea to reduce the domain gap between video and image datasets, this also introduces unfairness to the comparison to other self-supervised methods. This is because the filtering process with an ImageNet classifier implicitly takes advantage of ImageNet labels, and ImageNet labels are very effective on downstream image tasks as validated by the authors in Figure 4.

2. It is unsure how much the temporal deformations provided by a video dataset help with the pre-training VITO, which I think is one of the central questions for video contrastive learning for image downstream tasks. Because if temporal deformations are not useful, then why bother filtering a video dataset which costs much more than a image dataset. I am not sure if T=0 in Figure B.2 is a baseline that generate two views from the same frame in VideoNet. If so, this would be a helpful ablation that can be highlighted in the main text.

**Questions:**

I have a few questions on the experiment settings:

1. In Table 1, why are some results different from what are reported in their original papers? For example, VFS [8] on DAVIS is reported to be 68.9 in its paper, but 67.8 in this manuscript. Is this 67.8 number produced by the authors with a different experimental setting? Are most of numbers in Table 1 produced by the authors by fine-tuning on the publicly available checkpoints?

2. Why are the UCF-101 results in Table 1 significantly lower than other self-supervised learning algorithms on video? For example, in Feichtenhofer et al. [34] a ResNet-50 pre-trained on K400 can easily obtain an over 90% accuracy on UCF101?

3. The experiments on ImageNet-3DCC in Figure 2 are interesting. I am wondering why only evaluate MoCLR as the representative self-supervised algorithm, while not others like DINO?

4. I am wondering the training cost on VideoNet. For example, how many ImageNet-equivalent epochs have been used and how many crops are there in each step?


**Limitations:**

The authors adequately addressed the limitations potential negative societal impact of their work.

---

> ### Author Rebuttal · Authors · 2023-08-10
>
> We thank the reviewer for the comments and address the concerns below.
>
> **Weakness 1: VideoNet curation**
>
> We realize that the VideoNet curation procedure does involve utilizing the class distribution or implicit labels. However, most of the high-performance SSL methods we compare to were trained on ImageNet (which itself was manually curated) so we do not agree that there is anything unfair in our comparisons. Even the VINCE/CycleContrast video models leverage a similar curation strategy in construction of their R2V2 dataset.  Particularly, we outperform ImageNet pre-trained methods on video tasks, OOD robustness, and human alignment, demonstrating that we can leverage the benefits of ImageNet’s class diversity along with complex spatiotemporal deformations of objects to learn more general visual representations. However, to additionally alleviate concerns we have provided ablations of VITO using VideoNet and YT8M in Supp Table. B.1. We find that using these uncurated datasets, VITO outperforms all prior video-pretraining (including prior video pretraining leveraging the same datasets) and even outperforms an SSL baseline trained on the significantly larger, but uncurated JFT-300M image dataset.
>
> **Weakness 2: temporal ablations**
>
> We agree with the reviewer that assessing the impact of the temporal deformations is a central question. Indeed in our ablations in Fig B.2, T=0 is the spatial version of training where augmentations are only generated from individual frames. Therefore, we see a significant increase in performance including temporal deformations. We have now additionally shown the impact of temporal deformations through many more evaluations specifically emphasizing the impact of the temporal deformations on increased robustness (ImageNet 3dcc recognition) and human alignment on shape-biased tasks. For more information, see the global response and rebuttal PDF figures.
>
> **Question 1: DAVIS inconsistency**
>
> For fair comparison, we re-ran all the models we compare to by fine-tuning or evaluating linear classifiers (for ImageNet-based evals) on the publicly available checkpoints. Therefore, there are slight discrepancies in the DAVIS video segmentation numbers, but we believe all these comparisons are fair as they are all fine-tuned under the same conditions.
>
> **Question 2: UCF low numbers**
>
> Our UCF numbers are to the best of our knowledge, the best performance based on a pure image-based backbone. In fact, in Supp Table B.5 it can be seen that with simple temporal pooling strategies we surpass many prior video-based backbones and can get close to the Feichtenhofer numbers (under the same pre-training epochs). All prior work that show accuracies in the 90% range use training procedures that train with 3D ResNets, (2+1)D ResNets, or other architectures that allow for temporal integration in the model during training. We think it is significant that we can achieve numbers that are not too far from this just with spatial feature extraction and pooling of the output representations. In our new ablations (see global response), we see that this performance does not appear without the inclusion of temporal deformations during the training, indicating that there is implicit knowledge of the features relevant for defining action categories that can be captured in a purely image-based representation.
>
> **Question 3: ImageNet-3DCC**
>
> See our global response. We have now added DINO and many of the prior video pretraining works (cycle contrast, vince, vfs) and our ablations. We find that the strong robustness of VITO only appears with the combination of its methodological components, and is not replicated by other powerful SSL methods such as DINO or other video pretraining efforts.
>
> **Question 4: Computational budget**
>
> We designed the VITO training procedure such that it matched the computational budget of all of the models we compare to that are trained on ImageNet. In each step, for all models we train with 3 views and the 300 epochs refer to ImageNet-equivalent epochs (i.e. the same number of iterations needed for 300 ImageNet epochs). .

---

> > ### Author Response · Authors · 2023-08-21
> >
> > As the discussion period is coming to a close, we want to thank the reviewer again for their detailed review and ask if they have any other questions or concerns that have not been addressed by our rebuttal response.

---

### Official Review · Reviewer_3k9B · 2023-07-27

**Soundness:** 3 good
**Presentation:** 4 excellent
**Contribution:** 2 fair
**Rating:** 5
**Confidence:** 4

**Summary:**

The paper introduces a SSL method using video pre-training to produce general visual representations for both image and video tasks. The proposed VITO pipeline includes a data curation process and a video SSL technique based on MoCLR. The authors conduct a comprehensive evaluation of VITO's performance on diverse benchmarks, spanning detection, segmentation, video segmentation, classification, and out-of-distribution object recognition. Additionally, the paper includes a comparison of human-alignment on two benchmarks.

**Strengths:**

•	The paper is well-organized and easy to follow. The proposed components for the method, including data curation and various modifications on MoCLR, all make sense to me.

•	Ablation studies are conducted by the authors to validate the effectiveness of their proposed method.

**Weaknesses:**

1. The main method is based on the image SSL method MoCLR. While the authors demonstrate the effectiveness of multi-scale contrastive attention pooling, this component appears applicable to image SSL as well. Therefore, the technical novelty of applying an image SSL method to video datasets seems limited, especially considering the existence of other established video pre-training works.

2. The motivation behind choosing MoCLR as the starting baseline should be clarified. Given the availability of better contrastive learning methods and the superior performance of Masked-based pre-training (e.g., MAE), it would be beneficial to provide comparisons or discussions to justify this selection.

3. The data curation process plays a critical role in the overall pipeline. To gain more insights, additional ablation or analysis would be helpful.

3a.  For example, what are the the effect of the number of curated videos and different filtering strategies.

3b. For the ablation (Figure 4), considering that performance can be significantly influenced by pre-training data and downstream dataset distribution, it would be better to verify on more downstream tasks instead of only one dataset.

3c. It would be interesting to Investigate the scaling behavior on the training schedule, as 300 ImageNet epochs might be insufficient for large-scale video datasets.


4. In Table 1, incorporating more standard benchmarks would enhance the evaluation's comprehensiveness. For instance, Kinetics is a more widely adopted benchmark for video understanding compared to the small-scale UCF-101.

5. In B.5, VITO doesn’t show superior performance on image-based tasks, especially it has significant drop on ImageNet compared to ImageNet pre-trained methods. It might be worth to have some discussions around that as this is also important observations.

6. The paper lacks comparisons with some video-specific SSL methods, including contrastive learning-based and Masked-based methods (e.g., VideoMoCo, VideoMAE, MAE_ST, etc).

**Questions:**

Overall I think the video pre-training direction and results are interesting.
However, my main concerns revolve around the limited evaluation and results presentation, which lack sufficient ablation studies and comparisons, as mentioned in the weaknesses.

**Limitations:**

See weakness

---

> ### Author Rebuttal · Authors · 2023-08-10
>
> We thank the reviewer for their detailed comments and direct them to the global response for our summary. We will address each cited weakness point by point.
>
> **Weakness 1: technical novelty**
>
> We agree with the reviewer that the basic idea of applying an image-based SSL framework to a video dataset is not novel. In fact this has been attempted previously in the many cited works we compare to in Table 1 (CycleCon, VINCE, VFS, VIVI etc.). The novelty of our work lies in the careful methodological choices (for both the learning paradigm and dataset curation), which lead to significant empirical gains over all prior image-based video pretraining work. We concede that there are many successfully works which specifically train video-based architectures on video datasets, but these methods are severely limited in their application to general scene understanding, robust recognition, and comparisons to human behavior due to the fact that they cannot be adequately modified to maintain high performance on image-based tasks. We emphasize that a key aspect of our work is demonstrating how image-based representations trained on video data can be far more general: maintaining strong performance on spatiotemporal tasks while far outperforming on static image tasks.
>
> **Weakness 2: why MoCLR?**
>
> We will clarify this in the text, but we choose MoCLR based on it’s strong performance in [1] showing that it outperforms both contrastive (SimCLR) and non-contrastive (BYOL) formulations. We disagree that it is not a strong baseline. However, we have additionally compared to DINO in the robustness and alignment evaluations and find similar and sometimes worse behavior compared with MoCLR. We do not compare to all recent SoTA SSL methods such as MAE primarily because they use significantly larger architectures (generally transformers) that would not be architecture-matched and would obfuscate the current set of results (see general response for longer discussion).
>
> **Weaknesses 3a/3b: impact of curation is unclear**
>
> Regarding VideoNet data curation and ablations, we refer the reviewer to our general response. Briefly, we present ablations on the 4 scene understanding tasks using two other prominent video datasets (AudioSet and YT8M) (Supp. Table B1). While the performance drops slightly using alternative video datasets, we see that our method performs far better than prior image-based video pretraining efforts. VideoNet curation helps to close the gap with ImageNet pretraining but it is not necessary to outperform prior work. Nevertheless, we agree with the reviewer that more work should be done to understand how our results are impacted by different curation strategies. In particular, we believe a purely unsupervised curation approach may be possible in which we simply measure the similarity of our dataset and ImageNet images (without labels) in an unsupervised embedding space to better align the two, rather than using a classifier (similarly to DINO v2). We leave this for future work.
>
> **Weakness 3c: scaling epochs**
>
> We agree that investigating scaling behavior to longer epoch schedules is valuable and interesting. We did not have time to run further experiments on this, but will add text to cite this as future work.
>
> **Weakness 4: insufficiency of UCF101**
>
> UCF-101 is to our knowledge quite standard and widely accepted. While it may be easier than some benchmarks, many prior works do evaluate on it (See Supp Table B5). Additionally, there are very few prior works in image-based video pretraining which evaluate on Kinetics, which is the subset of prior work that we feel is the most important and direct comparison. Finally, many video-based pre-training in fact trains on the Kinetics dataset, so for those models, Kinetics is an in-distribution evaluation. Therefore, given these constraints, UCF presents a reasonable example of an OOD video benchmark. We also note that outside of the UCF benchmark, we additionally evaluate on video segmentation via DAVIS, which is a standard benchmark in the field.
>
> **Weakness 5: ImageNet numbers**
>
> The reviewer notes that VITO does not show superior performance on image-based tasks citing the ImageNet classification number. This is not true as VITO does outperform many ImageNet pretraining methods on tasks which use OOD datasets (segmentation, detection, and OOD recognition). The reason VITO does not outperform other methods on ImageNet validation accuracy is that this is an in-distribution evaluation for models trained on the clean ImageNet dataset. While we align the class distribution of VideoNet to that of ImageNet, the frames themselves are significantly more diverse and noisy meaning that ImageNet recognition is not strictly an “in-distribution” task for VideoNet pretraining, making the comparison unfair.
>
> **Weakness 6: comparing to video models**
>
> We provide some video-based comparisons in Supp Table B5 for comparing VITO on the UCF evaluation. However, on all image-based evaluations it is unclear how we can compare to these video-specific SSL methods as they all utilize video-specific architectures. It is well-known that these architectures cannot easily be adapted to handle image-based evaluations so we do not see an easy way to compare with these methods. This issue in fact exemplifies a benefit of VITO as it generalizes across image and video tasks seamlessly while maintaining strong performance in both domains.

---

> > ### Author Response · Authors · 2023-08-11
> > **quantitative comparisons with a masked image modeling approach**
> >
> > The reviewer requested comparisons with a masked image modeling baseline. While we had initially not found any such baselines which use a comparable architecture, in fact the recent work of Li et al. 2022 (A2-MIM [1]) develops a masked-image modeling methodology that can be applied to ResNet-50 architectures and is highly performant.
> >
> > To facilitate the discussion, we evaluated the A2-MIM models on multiple benchmarks. On the Geirhos human alignment benchmarks, we find it does not improve over standard supervised training in human alignment and still significantly underperforms VITO. On the ImageNet-3DCC benchmark, while A2MiM provides additional robustness (at the level of the stylized and robust resnets), it still underperforms VITO significantly. For detailed numbers see the tables below. We hope this provides an additional data point that demonstrates that VITO outperforms a variety of state-of-the-art SSL objectives in this respect.
> >
> > **Geirhos Human Alignment**
> >
> > | **Method**      | **Dataset** | **Accuracy difference (&#8595;)**   | **Observed. Consistency  (&#8593;)**  | **Ceiled Error Consistency ( &#8593;)**   |
> > | :---        |    :----:   |    :----: |    :----:   |    :----:  |
> > | VITO        |    VideoNet   |    **0.157** |   **0.564** | **0.422** |
> > | A2-MiM |    ImageNet   |    0.197   | 0.520  |  0.325  |
> > | DINO | ImageNet | 0.236 | 0.504 | 0.291 |
> >
> > **ImageNet-3dcc**
> > | **Method**      | **Dataset** | **$\Delta$ Accuracy Severity 1**   | **$\Delta$ Accuracy Severity 2**  | **$\Delta$ Accuracy Severity 3**   | **$\Delta$ Accuracy Severity 4**  | **$\Delta$ Accuracy Severity 5** |
> > | :---        |    :----:   |    :----: |    :----:   |    :----:  | :----:   |    :----:  |
> > | VITO        |    VideoNet   |    **-14.3** |   **-19.6** | **-24.8** |  **-29.6**  | **-34.1** |
> > | L2-Robust | ImageNet | -15.2 | -23.5 | -30.1 | -35.8 |  -40.5 |
> > | A2-MiM |    ImageNet   |  -16.3 |  -23.4 |  -30.5 |  -36.6 | -42.2 |
> > |DINO | ImageNet | -19.4 | -27.7 |  -34.7 |  -40.6 | -45.5 |
> >
> >
> > [1] Li, Siyuan, et al. "Architecture-Agnostic Masked Image Modeling--From ViT back to CNN." arXiv preprint arXiv:2205.13943 (2022).

---

> > > ### Comment · Reviewer_3k9B · 2023-08-19
> > > **Response to rebuttal**
> > >
> > > Thanks for the additional explanations and results. The rebuttal address some of my concerns while some of them are also left as future work. So I keep my borderline rating but update to borderline accept.

---

### Official Review · Reviewer_i1Wo · 2023-07-27

**Soundness:** 3 good
**Presentation:** 3 good
**Contribution:** 3 good
**Rating:** 6
**Confidence:** 3

**Summary:**

The paper proposes VITO, a new method for self-supervised video pretraining that learns general and human-aligned visual representations. It made several modifications over existing contrastive learning frameworks, including larger crop sizes, improved temporal sampling scheme, and multi-scale attention feature pooling for the projector. The paper also creates a video dataset (VideoNet) that aligns the class distribution with ImageNet, and partially redresses the imbalance between image and video learning. This improves spatial understanding compared to other video datasets.
Experiments are conducted on various tasks, including semantic segmentation / object detection on image, video object segmentation and classification, as well as OOD object recognition.

**Strengths:**

This paper is well-organized and easy to follow.

The technical contribution and the way to do data curation make sense to me.

The results shows that VITO matches or exceeds image pretraining on spatial tasks like object detection and segmentation while
outperforming other video pretraining methods. This shows it learns a general representation.

The ablation is comprehensive.



**Weaknesses:**

The human alignment evaluations are somewhat weak. Human alignment is a big area and whether saliency and shape bias tasks can serve as representative tasks for the evaluation concerns me. More comprehensive alignment benchmarks could be explored.

It would be great if the authors could also show baselines with the masked image modeling objective.

Overall I think this paper shows good results and presents things clearly. Thus my rating is weak accept. I encourage the authors to perform the above things I suggest to improve the quality of this paper further.

**Questions:**

Please see the weakness part.

---

> ### Author Rebuttal · Authors · 2023-08-10
>
> We thank the reviewer for their comments. Regarding the human alignment evaluations, we agree that more evaluations should be done; however, we focused on saliency and shape-bias tasks as these have been recent and prevalent tasks where many deep image models fail to capture important aspects of human behavior. In fact, the saliency benchmark that we evaluate on is an extremely large-scale study and we believe that this result is quite strong, given the fact that VITO provides greater saliency alignment than even the “harmonized” model (trained with an objective to explicitly match the human saliency maps). Additionally, shape vs texture bias has been a prevalent issue in the vision community given how strongly many models differ from humans. While the gap has closed for very large models trained on billions of images, our results demonstrate that even in a smaller scale setting, video pre-training may be a more natural solution. Nevertheless, more evaluations are always beneficial so if the reviewer has any suggestions for a preferable quantitative alignment benchmark, we are happy to evaluate and include this in the final paper.
>
> Regarding masked image modeling, we understand that this is a dominant recent SSL paradigm. However, much of the masked autoencoding (MAE) literature is focused on using transformer architectures, due to the need for operating on patchified inputs. Because we seek architecture-matched comparisons, this makes comparing to any masked image modeling objectives difficult. We have provided a brief scaling evaluation using Swin transformers in Supp Table B.4, but leave it to future work to do a thorough comparison across transformer architectures.

---

> > ### Author Response · Authors · 2023-08-11
> > **quantitative comparisons with masked image modeling approach**
> >
> > The reviewer requested comparisons with a masked image modeling baseline. While we had initially not found any such baselines which use a comparable architecture, in fact the recent work of Li et al. 2022 (A2-MIM [1]) develops a masked-image modeling methodology that can be applied to ResNet-50 architectures and is highly performant.
> >
> > To facilitate the discussion, we evaluated the A2-MIM models on multiple benchmarks. On the Geirhos human alignment benchmarks, we find it does not improve over standard supervised training in human alignment and still significantly underperforms VITO. On the ImageNet-3DCC benchmark, while A2MiM provides additional robustness (at the level of the stylized and robust resnets), it still underperforms VITO significantly. For detailed numbers see the tables below. We hope this provides an additional data point that demonstrates that VITO outperforms a variety of state-of-the-art SSL objectives in this respect.
> >
> > **Geirhos Human Alignment**
> >
> > | **Method**      | **Dataset** | **Accuracy difference (&#8595;)**   | **Observed. Consistency  (&#8593;)**  | **Ceiled Error Consistency ( &#8593;)**   |
> > | :---        |    :----:   |    :----: |    :----:   |    :----:  |
> > | VITO        |    VideoNet   |    **0.157** |   **0.564** | **0.422** |
> > | A2-MiM |    ImageNet   |    0.197   | 0.520  |  0.325  |
> > | DINO | ImageNet | 0.236 | 0.504 | 0.291 |
> >
> > **ImageNet-3dcc**
> > | **Method**      | **Dataset** | **$\Delta$ Accuracy Severity 1**   | **$\Delta$ Accuracy Severity 2**  | **$\Delta$ Accuracy Severity 3**   | **$\Delta$ Accuracy Severity 4**  | **$\Delta$ Accuracy Severity 5** |
> > | :---        |    :----:   |    :----: |    :----:   |    :----:  | :----:   |    :----:  |
> > | VITO        |    VideoNet   |    **-14.3** |   **-19.6** | **-24.8** |  **-29.6**  | **-34.1** |
> > | L2-Robust | ImageNet | -15.2 | -23.5 | -30.1 | -35.8 |  -40.5 |
> > | A2-MiM |    ImageNet   |  -16.3 |  -23.4 |  -30.5 |  -36.6 | -42.2 |
> > |DINO | ImageNet | -19.4 | -27.7 |  -34.7 |  -40.6 | -45.5 |
> >
> >
> > [1] Li, Siyuan, et al. "Architecture-Agnostic Masked Image Modeling--From ViT back to CNN." arXiv preprint arXiv:2205.13943 (2022).

---

> > > ### Comment · Reviewer_i1Wo · 2023-08-18
> > >
> > > I appreciate the authors considering changing the title and the newly added results. I will keep my rating as weak accept.

---

### Author Rebuttal · Authors · 2023-08-10

We thank all reviewers for the feedback and acknowledging our work's clear presentation of a novel step towards learning more general human-aligned visual models. We will address global concerns here and individual comments separately. Please see the rebuttal PDF for new results.

**Ablations:** Reviewers emphasized the need for more evaluations on ablated models to justify the critical components of our final model. To answer these questions, we present new results evaluating multiple ablations of VITO along different dimensions: UCF101 action recognition, OOD image recognition (ImageNet-A/ImageNet-vid), recognition under natural corruption (ImageNet 3dcc) and human alignment (shape bias tasks). We evaluated the following models on these tasks, seen in Table 1, Table 2,  and Figure 3 (right panel):
* MoCLR VideoNet: MoCLR applied to VideoNet frames, no temp. augmentations.
* VITO spatial: 2-scale attention pooling, no temp. augmentations.
* VITO (1scale w/o attn): All augmentations, 1-scale embedding, no attention pooling.
* VITO (2scale w/o attn): 2-scale embeddings, all augmentations, no attention pooling.
* VITO (1scale): Full VITO, 1-scale embedding.
* VITO (AudioSet): Full VITO, pretraining data: uncurated AudioSet.

These models exemplify the key method elements (spatial vs spatio-temporal deformations, multi-scale contrastive attention, and VideoNet dataset).

_Rebuttal Table 1_ data is consistent with the more limited ablations in App Fig. B2 and App Table B.1. First, incorporating temporal deformations contributes greatly to the best model performance (temporal VideoNet models outperform the VITO-spatial and MoCLR models on all benchmarks). Second, architecturally, the multi-scale aspect is the most significant component on its own, but when combined with attention pooling there is a synergistic effect that leads to significant improvements, especially on OOD recognition benchmarks. Moreover, VideoNet models outperform those trained on other datasets such as Audioset. Along with the ablations in App. Table B.1, this suggests that the importance of VideoNet is in shaping the class distribution rather than just having large visual diversity.

_Rebuttal Table 2_ shows that the full VITO method produces the best human alignment in terms of ceiled error consistency. Notably, other high-performing models are _all VITO ablations that were trained with temporal deformations_. This surprisingly includes the AudioSet pretrained model, even though it has poorer performance on the computer vision tasks. This result is quite striking, as previous image-based video pretraining methods perform quite poorly (VINCE [7], VFS [8], and CycleCon [9]). This suggests that our specific methodology better leverages the spatio-temporal data to produce learned representations that are more human-aligned. As suggested, we now include a comparison with CLIP, which does have better error-consistency than our best model. However, CLIP is trained with large-scale language supervision (400M image-text pairs), and is thus not directly comparable. More relevant are comparisons to image- and video-only SSL methods, where VITO significantly outperforms prior work.

_Rebuttal Fig 3_ (right panel): In recognition under natural corruptions (ImageNet-3DCC), VITO outperforms all ablated versions. Again, there is a striking gap between models trained with spatio-temporal deformations, which are far more robust than those trained only with spatial augmentations. Coupled with the alignment results, this strongly highlights the importance of video vs. image pretraining.

**Comparisons to MAEs and transformer-based architectures:** While we agree with the reviewers that transformer architectures would greatly benefit the work, this would require a significant replication effort that we leave to future work. It would be unfair to just compare ResNet-50 models with transformers due to the vast differences in expressive power. To reiterate, the goal of this work is to _compare image representations learned from videos with those learned from images, assuming matched model capacities_. In addition, some of our most important comparisons are with prior work that also attempted to learn image representations from video data. These also mostly performed experiments with ResNet-50s, informing our initial architecture choice to have apples-to-apples comparisons with prior work.

**Comparisons to SoTA video models**: Most SoTA video-based methods utilize specialized video architectures, and thus cannot be evaluated on general image-based tasks such as scene understanding, robust recognition, and human alignment. As a result, while these methods perform well on video tasks such as action recognition, they remain less general than ours and are less appealing candidates for robust, human-aligned visual representations.

Lastly, we wish to clarify potential confusion over our title and introduction. We do not mean to claim we have learned the most human-aligned, general visual representation. Yet, because most of the current literature in this space is solely focused on achieving more general representations via scale (larger image datasets and models), we wanted to probe alternative ways to achieve this goal. Our work shows that video pretraining can yield more general, robust, and human-aligned visual representations by leveraging natural spatio-temporal deformations, even at a relatively small scale. This contribution can be complementary with scaling to larger architectures, datasets, and sources of supervision (e.g. language alignment). As an example, recent work has shown the benefits of combining image-level self-supervision and multimodal image-text alignment [Mu]. Naturally, combining our work in self-supervised video pretraining with multimodal image-text alignment appears as a potentially promising direction for future work.
[Mu] Mu, Norman, et al. "Slip: Self-supervision meets language-image pretraining."

---

### Decision · Program_Chairs · 2023-09-21

**Decision:**

Accept (poster)

**Comment:**

This paper proposes a new method for self-supervised video pretraining that learns general and human-aligned visual representations. The results are quite compelling and the reviewers seem to generally agree there is a solid methodological contribution to the field. Reviewers raise points about limited ablation, but the additional points brought up in the rebuttal help in this regard.  What is most striking is that the method seems to improve robustness and be competitive with models specifically trained to combat corruption robustness, leading some credence to the main motivation of the paper.

Thus the decision will be to accept.